# Martian magmatism from plume metasomatized mantle

James M.D. Day [1], Kimberly T. Tait[2], Arya Udry[3], Frédéric Moynier[4], Yang Liu[5] & Clive R. Neal[6]

Direct analysis of the composition of Mars is possible through delivery of meteorites to Earth. Martian meteorites include ~165 to 2400 Ma shergottites, originating from depleted to enriched mantle sources, and ~1340 Ma nakhlites and chassignites, formed by low degree partial melting of a depleted mantle source. To date, no unified model has been proposed to explain the petrogenesis of these distinct rock types, despite their importance for understanding the formation and evolution of Mars. Here we report a coherent geochemical dataset for shergottites, nakhlites and chassignites revealing fundamental differences in sources. Shergottites have lower Nb/Y at a given Zr/Y than nakhlites or chassignites, a relationship nearly identical to terrestrial Hawaiian main shield and rejuvenated volcanism. Nakhlite and chassignite compositions are consistent with melting of hydrated and metasomatized depleted mantle lithosphere, whereas shergottite melts originate from deep mantle sources. Generation of martian magmas can be explained by temporally distinct melting episodes within and below dynamically supported and variably metasomatized lithosphere, by long-lived, static mantle plumes.

[1] Scripps Institution of Oceanography, University of California San Diego, La Jolla, CA 92093, USA. [2] Department of Natural History, Royal Ontario Museum, Toronto, ON M5S 2C6, Canada. [3] Department of Geoscience, University of Nevada Las Vegas, Las Vegas, NV 89154, USA. [4] Institut de Physique du Globe de Paris, Université Sorbonne Paris Cité, Université Paris Diderot, 1 Rue Jussieu, 75328 Paris cedex 05, France. [5] Jet Propulsion Laboratory, California Institute of Technology, Pasadena, CA 91109, USA. [6] Department of Civil and Environmental Engineering and Earth Science, University of Notre Dame, Notre Dame, IN 46556, USA. Correspondence and requests for materials should be addressed to J.M.D.D. (email: jmdday@ucsd.edu)

Comparative planetology relies on assumptions that physical processes occurring for Earth can be used to compare with processes acting on other planetary bodies. This logic has been applied to Mars, where the broadly basaltic surface composition has been considered analogous to volcanism in terrestrial hotspot locations, including Hawaii[1]. Large volcanic edifices on Mars, including Olympus Mons, Alba Patera, Elysium, Arsia, Pavonis, and Ascraeus have been attributed to partial melting above mantle plumes[2,3]. Mars does not appear to have had plate tectonics, implicating long-lived and static volcanism above mantle plumes, in contrast to age-progressive volcanism occurring in oceanic and continental settings on Earth[4,5]. This difference, coupled with the distinct thickness and flexural rigidity of the elastic lithosphere on both planets, make assumptions of similar volcano-magmatic processes challenging. For example, observed pre-shield, shield, post-shield erosional, and rejuvenated stages of volcanism observed in Hawaii[6] have not been recognized for Mars owing to the lack of plate motion.

Direct and precise geochemical observations for Mars are possible by analysis of meteorites recognized as having a martian heritage, from trapped Ar, Xe, and $N_2$ signatures identical to the martian atmosphere[7–9], and by their distinct oxygen isotope compositions[10]. These meteorites, which represent near-surface extrusive or hypabyssal intrusive rocks within the martian crust, including shergottites, nakhlites and chassignites, and ancient (4100–4400 Ma) crustal rocks (ALH 84001; NWA 7034/7533 and their pairs), are the ground truth for remote sensing[11,12]. Shergottites are the most geochemically diverse group of martian meteorites. They span a range of compositions, with evidence for long-term incompatible trace-element-depleted and trace-element-enriched reservoirs in Mars that were formed as early as ~4504 Ma[13,14]. Originally, the incompatible-element-depleted reservoir was considered to represent the mantle and, the incompatible-element-enriched reservoir, the crust[15]. This straightforward interpretation has been increasingly challenged by petrological, geochemical, and isotopic data, indicating the presence of both incompatible-element-depleted and incompatible-element-enriched martian mantle reservoirs[16–19].

The geochemical identity of shergottite source reservoirs is well expressed in $^{87}Sr/^{86}Sr$–$^{143}Nd/^{144}Nd$ space, where samples are corrected for ingrowth from $^{87}Rb$ and $^{147}Sm$ decay since crystallization (Fig. 1). Shergottites define long-term Rb and Nd-depleted, intermediate, and -enriched groups and, collectively, span compositions a factor of four greater than found in terrestrial ocean island or mid-ocean ridge basalts. The 1270–1420 Ma (average of 1340 ± 40 Ma) nakhlites and chassignites[20] are isotopically distinct from shergottites, with depleted Sr and Nd isotope compositions. Nakhlites (clinopyroxene-rich mafic rocks) and chassignites (dunites) are petrogenetically associated low-degree partial melts from the same depleted mantle source[20–22]. The relationship of nakhlites and chassignites to shergottites is poorly understood. Both depleted shergottites and nakhlites have $^{142}Nd$ enrichments consistent with early-formed Sm/Nd-enriched mantle sources[23,24], but $^{182}W$ and $^{142}Nd$ data for nakhlites indicate a more complex origin than for shergottites that cannot be explained by early silicate differentiation alone[14]. Instead, late incompatible-element source enrichment must have occurred between major silicate differentiation at 4504 Ma and crystallization of the nakhlites at ~1340 Ma[14].

In this study, we present a coherent geochemical dataset for 24 shergottites and 16 nakhlites that highlights fundamental differences in their incompatible trace-element compositions. These trace-element differences can be explained by melting of metasomatized mantle sources, likely the martian lithosphere, for nakhlites and chassignites, and deep mantle plume sources for shergottites. Combined with Sr-Nd isotope variations and

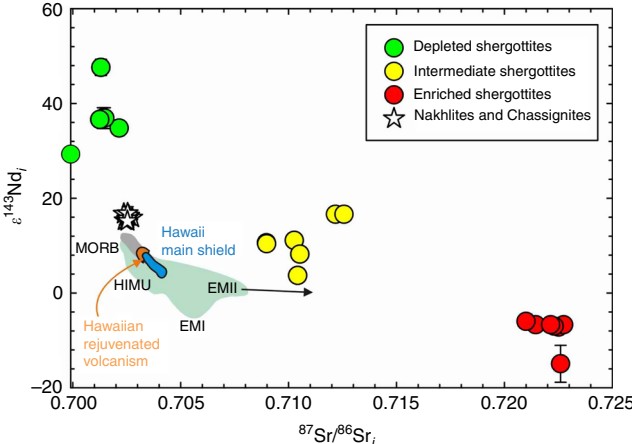

**Fig. 1** Comparison of the strontium and neodymium isotopic compositions of basaltic volcanic rocks at the time of their crystallization (i = initial crystallization time) from Mars and Earth. Terrestrial basalts include ocean island basalts (green field), mid-ocean ridge basalts (MORB; gray field), Hawaiian Main Shield (Mauna Loa, Mauna Kea; blue field), and Hawaiian Rejuvenated Volcanism (Honolulu volcanics, North Arch, Niihau, Kaula; orange field). Approximate terrestrial "HIMU" (high$^{238}U/^{204}Pb$), EMI and EMII (enriched mantle I and II) geochemical reservoirs are shown. $\varepsilon^{143}Nd_i$ = ([($^{143}Nd/^{144}Nd_i)_{sample}/(^{143}Nd/^{144}Nd_i)_{chondritic}]$−1) × $10^4$. Data sources are given in the Methods section and error bars are smaller than symbols unless shown

geophysical evidence for high effective elastic thickness of the martian lithosphere (≥150 km), these data are consistent with generation of shergottites from mantle plume partial melts and nakhlites and chassignites from lower degree partial melting of hydrated mantle during loading of the martian lithosphere. These modes of formation for martian meteorites are analogous to Hawaiian main shield and rejuvenated volcanism, respectively. Instead of loading of moving oceanic lithosphere above a stationary hotspot, however, Mars manifests dynamic support of thick lithosphere by static mantle plumes, and the generation of large volcanic loads and the largest volcanoes in the Solar System.

## Results

**Compositions of shergottites and nakhlites.** We present new data for 24 shergottite meteorites and compare these data with those obtained for 16 nakhlites and chassignites using identical analytical protocols that involved complete dissolution of samples (see Methods). Measurement under identical analytical conditions removes non-systematic error from analysis of individual meteorites in different laboratories. Shergottites range from picrobasalt to basaltic compositions, with enriched shergottites having elevated total alkalis ($Na_2O + K_2O$) relative to intermediate or depleted shergottites (Fig. 2). Nakhlites and chassignites span a wide range of compositions from low-$SiO_2$ picrobasalt to basaltic andesite. Shergottites range from incompatible trace-element-depleted to trace-element-enriched compositions that are distinct from nakhlites and chassignites and are well expressed when comparing incompatible trace-element ratios (Ce/Pb, Nb/Y, La/Yb, Ba/Nb, Zr/Ti, and Zr/Nb) vs. indices of magmatic fractionation (MgO; Fig. 3).

Depleted shergottites have lower absolute abundances of incompatible trace elements and rare earth elements (REEs) than intermediate or enriched shergottites and low CI-chondrite-normalized light REE/heavy REE (La/Yb$_n$ = 0.114 ± 0.044; 1 SD), (Fig. 4). Ratios of La/Yb for intermediate shergottites (La/Yb$_n$ =

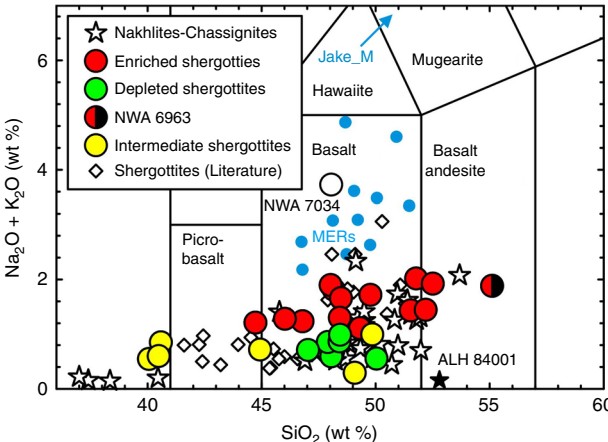

**Fig. 2** Total alkalis (Na$_2$O + K$_2$O) vs. silica diagram for martian meteorites and data for rocks from the Mars Exploration Rovers. Blue dots are Mars Exploration Rover data (MERs; ref. [13]). Chassignite meteorites are star symbols with <41wt % SiO$_2$. Data sources are this study and literature outlined in the Methods section. Analytical uncertainties are smaller than symbols

0.39 ± 0.10) are lower than for enriched shergottites (La/Yb$_n$ = 0.930 ± 0.078). In comparison, nakhlites and chassignites have a wider range of La/Yb$_n$ = 0.99 ± 0.28, while NWA 6963 has La/Yb$_n$ of 4.9. Hot desert meteorites have been variably affected by terrestrial alteration for Ba, Sr, and U, and NWA 2986 has a prominent Pb anomaly, which may also be due to terrestrial contamination. Sample NWA 6963 appears to be exceptional in that it has the texture and major-element geochemical composition of a differentiated shergottite (Fig. 2), but the incompatible trace-element composition of a nakhlite (Fig. 4).

**Distinct martian geochemical characteristics.** Major-element and trace-element results for shergottites, nakhlites, and chassignites reinforce several previously identified characteristics of these rocks. Shergottite data highlights the incompatible-element-depleted, intermediate and -enriched compositions of samples[11]. Intermediate shergottites are the most mafic analyzed in this study, with enriched shergottites generally being more silica rich. Nakhlites and chassignites have light REE enrichment relative to the heavy REE[20]. The data also reveal fundamental differences in the geochemistry of shergottites vs. nakhlites and chassignites. Despite having similar Zr/Hf ratios (shergottites = 29.7 ± 4.5; nakhlites and chassignites = 30.6 ± 1.7), fractionation of Zr and Hf in shergottites is mirrored by nakhlites and chassignites. For example, calculated Zr* (Zr$_n$/√[Nd$_n$ × Sm$_n$], where $n$ is the normalized value to CI-chondrite) or Hf* (Hf$_n$/√[Nd$_n$ × Sm$_n$]) are distinct between shergottites (Zr* = 1.6 ± 0.3; Hf* = 2.0 ± 0.4) and nakhlites and chassignites (Zr* = 0.4 ± 0.1; Hf* = 0.5 ± 0.1). Similar differences also exist for Nb (Nb* = Nb$_n$/√[La$_n$ × Th$_n$]) and Ta (Ta* (Ta$_n$/√[La$_n$ × Th$_n$]) between shergottites (Nb* = 1.4 ± 0.3; Ta* = 1.3 ± 0.3) vs. nakhlites and chassignites (Nb* = 0.8 ± 0.1; Ta* = 0.9 ± 0.1). As expected for melts from depleted mantle sources, nakhlite and chassignites have generally lower absolute abundances of Nb, Y, Zr, and Ta than intermediate or enriched shergottites, but higher relative abundances than the depleted shergottites.

## Discussion
The available samples for study from Mars are meteorites, many of which are finds and resided on Earth for significant periods of time (>100 years) prior to collection. The meteorites also have limited total mass making analysis of large (>30 g) well-homogenized aliquots, as is typically done for terrestrial igneous rocks, virtually impossible. Many of the martian meteorites that we studied show evidence for terrestrial alteration effects, including anomalously elevated large ion lithophile element (e.g., Cs, Ba, Sr) and U abundances. For example, the depleted shergottite DaG 476—a hot desert find—has Ba/Nb of ~1500, relative to Antarctic martian meteorite finds with Ba/Nb typically <30. Terrestrial alteration has had no effect on the majority of the REEs or high field strength elements (HFSEs) such as Nb, Ta, Hf, Zr, or Y. This is reflected in well-defined groupings for nakhlites and chassignites, and for incompatible element-enriched, intermediate, and -depleted shergottites for Nb/Y, La/Yb, and Zr/Nb (Fig. 3). For example, we find no systematic differences in the REE or HFSE inter-element ratios for the different geochemical types of shergottite, or between nakhlites and chassignites, from either hot or cold deserts.

The mode effect—where non-representative volumes of rock are chosen, resulting in greater variability in chemical measurements—is a significant issue if not recognized in meteorites owing to the small sample sizes typically studied by investigators (generally <5 g). To examine this effect, we analyzed separate aliquots of material from shergotittes NWA 7042, NWA 5298, NWA 7257, NWA 3171, DaG 476, and Tissint, and for nakhlites NWA 998, Nakhla, and MIL 090136. We find that while variations in absolute abundances of elements for separate aliquots can be large, up to 20% in some cases, the relative abundances and inter-element ratio variations are more limited (typically «10%). Occasionally, there are large variations in major-element abundances; for example, 50% variation for K$_2$O in NWA 7042, and extreme heterogeneity between separate aliquots (e.g., NWA 998). Despite these variations, the inter-element ratios of key elements (e.g., Nb, Zr, Ta, Hf, Y) and the REE patterns of samples are unaffected, allowing assessment of the magmatic processes acting in Mars.

A remarkable aspect of shergottite, nakhlite, and chassignite compositions is their similarity to geochemical differences observed between Hawaii main shield-stage and rejuvenated-stage volcanic rocks. Hawaiian shield-stage lavas form from the highest degrees of partial melting, above the plume conduit[25]. Rejuvenated-stage lavas occur after the main stages of volcanism and erupted peripheral to the Hawaiian plume center[26–30]. These rejuvenated alkalic basaltic lavas are incompatible-element-enriched low-degree partial melts, yet have Sr and Nd isotopic compositions requiring long-term depleted mantle sources relative to the shield source. These characteristics are like those for nakhlites and chassignites (Fig. 5). Such characteristics are difficult to explain by crustal contamination of parental magmas, since martian regolith (represented by NWA 7034/7533 and its pairs) is similar to terrestrial crust in having lower Nb/Y than nakhlites and chassignites. The variations in Nb/Y and Zr/Y for nakhlites are therefore common to the sense of fractionation for Hawaiian rejuvenated lavas and cannot be produced by crustal contamination.

A hallmark of Hawaiian rejuvenated-stage volcanism is that it begins after a volcanically quiescent period, requiring a mechanism to engender further melting. Several models have been proposed[28], with the most popular invoking the depleted source as an intrinsic part of the Hawaiian plume that has been pervasively metasomatized by infinitesimally low-degree melts[28–30]. Maximum extents of rejuvenated melting occur during decompression of the mantle by lithospheric flexure, reaching a maximum ~200 km downstream from the plume[30]. For Mars, joint analysis of gravity and topography has been used to estimate effective elastic thickness of the lithosphere, typically interpreted

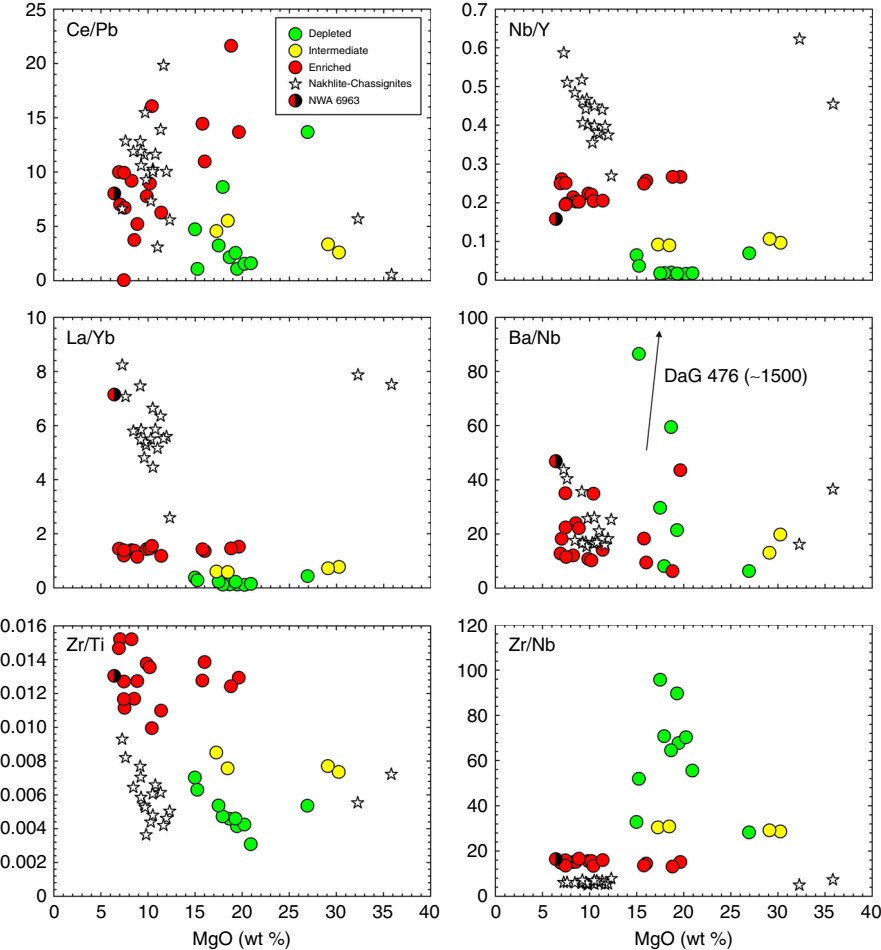

**Fig. 3** MgO vs. Ce/Pb, Nb/Y, La/Yb, Ba/Nb, Zr/Ti, and Zr/Nb for shergottites, nakhlites, and chassignites. Chassignite meteorites are star symbols with >30 wt % MgO. Data are from this study and analytical uncertainties are smaller than symbols

as the depth of the 650 °C isotherm, or the point at which the material is too weak to support geological stresses over >$10^8$ years. It has been shown that thickness of the martian lithosphere increases with age and decreases with radiogenic power from K, Th, and U decay[2]. In detail, the region with the most massive magmatism—Tharsis—appears to be dynamically supported, suggesting the presence of a mantle plume[2,3], and consistent with relatively young (<100 Ma) volcanism in this region[31]. It has also been proposed that subsurface variations in load, including the presence of a depleted mantle composition, might exist underneath nearly all large martian volcanoes[3].

We interpret shergottites to be analogous to shield-stage or pre-shield-stage and post-shield-stage, whereas nakhlites and chassignites are analogous to rejuvenated-stage igneous rocks. In the absence of plate tectonics, stationary plume-generated melting would be expected to strongly deplete portions of the martian lithosphere and mantle, as has been suggested from gravity and topography[3]. During plume impingement and maturation, eruption of large volumes of basaltic magmas would occur leading to loading of the lithosphere, a flexural bulge and a flexural moat around the volcanic load[32]. Flexural moats are observed around nearly all the Tharsis volcanoes from gravity data[33] and are similar to, although significantly larger than, flexural bulges around Hawaiian-Emperor chain volcanoes[34]. To generate partial melts responsible for nakhlites and chassignites by lithospheric flexure, metasomatism of portions of depleted mantle would be

required. Evidence for a water-bearing lithosphere is provided from its rheology[2], and from addition of fluids into nakhlite and chassignites[20,22].

Niobium/Y-Zr/Y systematics of nakhlites and chassignites support a model of partial melting of depleted mantle metasomatized by fluids. The relationships of high Nb/Y of Hawaiian rejuvenated and main shield lavas and nakhlites and chassignites and shergottites must reflect fundamental differences in Nb, Y, and Zr behavior during partial melting. Phases that have affinity for Nb, but less so for Zr or Y, include amphibole, rutile, ilmenite, and mica, of which mica and amphibole have been observed in nakhlite and chassignite melt inclusions[22]. Complete exhaustion of such phases during melting would explain the high Nb/Y of both nakhlite and chassignites and Hawaiian rejuvenated lavas. Models of 5–10% partial melting of primitive mantle reproduce Hawaiian shield compositions, whereas those of depleted mantle reproduce mid-ocean ridge basalt compositions (Fig. 5). Low-degree partial melts of a primitive mantle source could, in theory, generate Hawaiian rejuvenated lavas, but this is inconsistent with their depleted source character from Sr-Nd isotope systematics. The presence and complete exhaustion of limited quantities of hydrous phases (amphibole, mica) and rutile can generate high Nb/Y melts such as nakhlite and chassignites or Hawaiian rejuvenated volcanism.

The similarity between martian meteorites and terrestrial basaltic analogs implies important planetary comparisons. In

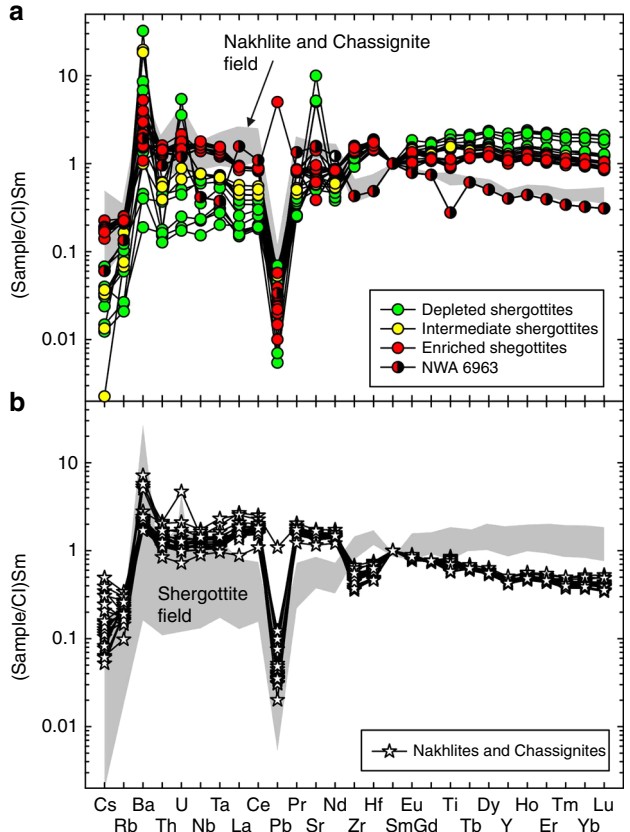

**Fig. 4** Double-normalized (to CI-chondrite and Sm) incompatible trace-element plots for martian shergottites, and nakhlites and chassignites. Shergottites are shown in **a** and nakhlites and chassignites are shown in **b**. Shergottite NWA 6963 is highlighted due to its similarity in incompatible element composition to nakhlites. Normalization to CI-chondrite from ref. [39]. Analytical uncertainties are smaller than symbols

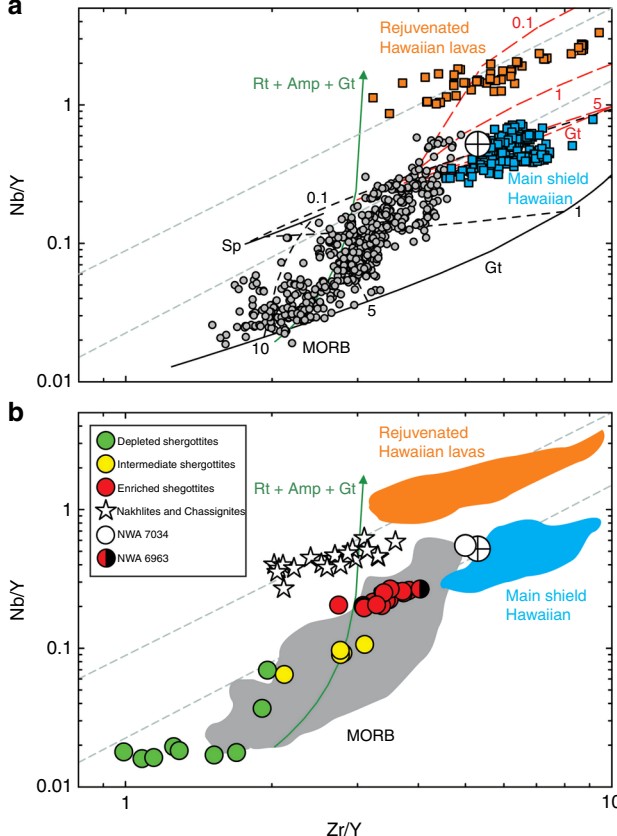

**Fig. 5** Partial melting processes in Mars and Earth. Niobium–zirconium–yttrium discrimination diagrams showing **a** terrestrial mid-ocean ridge basalts (MORB) and Hawaiian main shield building and rejuvenated phase lavas. Partial melt models show melting of a terrestrial depleted mantle source and a metasomatized mantle source, containing amphibole and rutile. Gray dashed lines denote envelope of total variation seen in Icelandic samples using this type of plot. Lines in red are a primitive mantle model, lines in black are a depleted mantle model, and the green line is the exhaustion vector for rutile (Rt), amphibole (Amp), and garnet (Gt) in a metasomatized martian mantle source. Numbers (0.1, 1, 5, 10) associated with dashed tie lines between pure garnet and spinel endmembers are partial melt increments in percent, and Sp is spinel. **b** Comparison of martian meteorites with terrestrial lavas. Crossed circle is the terrestrial continental crust composition. Model parameters and data sources are given in the Methods section and analytical uncertainties are smaller than symbols

detail, however, differences in isotopic compositions would suggest that the homogenization of mantle isotopic compositions by convective mixing has not been as effective for Mars as for Earth. The variation in Sr-Nd isotope systematics for martian meteorites, and the ~4500 Ma large-scale differentiation ages from these isotope systems[14,35], indicate limited mantle stirring since planetary differentiation. This tectonic difference also explains the limited long-term time-integrated variation between Hawaiian main shield and rejuvenated lavas compared to the large difference in isotopic depletion for nakhlites and chassignites relative to intermediate or enriched shergottites.

Our interpretation of nakhlites and chassignites as forming during loading of the martian lithosphere provides predictive power. Martian rejuvenated volcanism should occur where lithosphere thickened and stabilized and where flexural uplift engendered partial melting. Sources of nakhlites and chassignites should be peripheral to plume volcanism at ~1300 to 1400 Ma and perhaps over longer periods of time for martian igneous rocks not currently sampled in the terrestrial meteorite collection. Additionally, shergottite crystallization ages (165–2400 Ma) generally preclude a direct link between nakhlites and chassignites. Nakhlites and chassignites come from a source that had prior melt depletion, and the previously extracted melts were likely similar to shergottites to explain mirrored elemental abundance patterns. Observations of 2400 to ~160 Ma shergottite magmatism on Mars supports continued and persistent plume magmatism for at least two billion years[13,36]. This link may be further strengthened by the

observation that NWA 6963—a shergottite—has trace-element abundances like nakhlites, suggesting some geochemical affinities between magma types. We interpret the current suite of martian meteorites to represent at least two phases of plume-related magmatism. The distinctive nature of martian meteorites to remotely sensed martian surface samples, in particular the higher alkali contents of some Gusev and Gale crater samples[37,38], possibly implies that alkali volcanism on Mars is driven by low-degree partial melting from metasomatism of martian lithosphere in response to mantle plume impingement, offering a mechanism for explaining the apparent differences observed between meteorites and mission data for Mars.

## Methods
**Sample preparation and analytical methods**. New data were obtained for 24 shergottite meteorites, spanning the known range of incompatible element

depletion and enrichment and showing a range of MgO contents, from 6.4 to 29 wt %. These data are compared with those obtained for 16 nakhlites and chassignites using identical protocols (Supplementary Data 1) and that were previously reported by Udry and Day[20]. For the desert finds, a Wells low-loss diamond wire saw was used to access material away from the fusion crust surface. For all materials, weathering or fusion crust surfaces were removed by crushing sample fragments with limited force, or by sawing, with sawn surfaces being thoroughly sanded with corundum paper, prior to generation of fine ground sample powders that were prepared using a clean alumina mortar and pestle.

Analytical procedures were undertaken at the Scripps Isotope Geochemistry Laboratory (SIGL). The standard technique used for all samples was digestion of 50 mg of sample powder in Teflon-distilled 27.5 M HF (4 mL) and 15.7 M $HNO_3$ (1 mL) for >72 h on a hotplate at 150 °C, along with total procedural blanks and terrestrial basalt and andesite standards (BHVO-2, BCR-2, BIR-1a, AGV-2). The 50 mg powder aliquot was taken from a larger homogenized sample powder >0.5 g (up to 2 g) in all cases. Acid attack led to complete dissolution of rock samples, generating clear solutions, with no remaining solid material. Samples were sequentially dried and taken up in concentrated $HNO_3$ to destroy fluorides, followed by doping with indium to monitor instrumental drift during analysis, and then diluted to a factor of 5000. Trace-element abundances were determined using a ThermoScientific iCAP Qc quadrupole inductively coupled plasma mass spectrometer (ICP-MS) and all data are blank-corrected. Long-term reproducibility of abundance data, based on the 3-year analytical campaign is better than 6% for most elements, except for Mo, Te, and Se (>10%).

Major-element abundances were determined on the same solutions used for trace-element abundance determination, using a 50,000× dilution factor. Major-element abundances were determined using a ThermoScientific iCAP Qc quadropole ICP-MS. For major elements, Si was derived by difference, with reproducibility of other elements measured on the BHVO-2 reference material being better than 3%, except $Na_2O$ (7.1%). Tait and Day[19] measured ocean island basalt samples using both X-ray fluorescence and solution ICP-MS techniques, finding strong agreement between the two methods for MgO, $Na_2O$, $TiO_2$, $Al_2O_3$, and $P_2O_5$, and reasonable agreement for FeO, MnO, CaO, $K_2O$, and $SiO_2$. They also obtained precision typically better than 5% and accuracy from the accepted values of the BCR-2, BIR-1g and AGV-2 standard reference materials of better than 6%. The ICP-MS method is ideal for determination of major and trace elements in fresh samples with minimal alteration for which limited sample mass (<2 g) is available for study, such as meteorites. A caveat is that limited sample masses can lead to a mode effect, where non-representative volumes of rock are chosen, resulting in greater variability in chemical measurements, as described in the Discussion section.

**Complete sample dissolution**. The procedures of digestion described above have been successfully and routinely used to fully dissolve terrestrial mafic igneous rocks[39]. For example, the method completely dissolves mafic igneous rock samples, like shergottites and nakhlites and chassignites. To confirm these results, we analyzed some of the standard reference materials and one martian meteorite (Nakhla) by first digesting the samples using a "Paar-Bomb digestion". Identical protocols to the method described above were used, except that these materials were loaded into Teflon sleeves surrounded by a steel pressure vessel and heated to 180 °C in an oven for 72 h, and major-element abundances were not measured. As with the standard method of digestion, complete dissolution was accomplished. There are no observable differences, outside analytical uncertainties, or mode effect issues (e.g., replicate measurements of different fragments of Nakhla), for ratios of Zr/Y or Nb/Y, with the standard reference materials, BHVO-2 and BCR-2, reproducing within 2% for these ratios. Furthermore, a literature survey of martian shergottite, nakhlite, and chassignite meteorite trace-element abundance data show broad similarity with the results for the HFSE and REE presented here. The distinct Zr/Y, Nb/Y, Nb*, Hf*, Zr*, Ta*, and Nb* of nakhlites and chassignites vs. shergottites reflect true variations between the meteorites, rather than any analytical artifacts or terrestrial alteration effects.

**Data sources for Figs. 1, 5**. Age-corrected Sr and Nd isotope data for shergottites, nakhlites, and chassignites in Fig. 1 were obtained from refs. [13,35,36,40–50] and references therein. Terrestrial ocean island basalt and mid-ocean ridge basalt Sr-Nd isotope data for Fig. 1 were obtained from refs. [25–30,39,51–59] and references therein. Since these data are from a range of laboratory sources, correction for non-systematic laboratory errors is not possible. Data for Fig. 5 are from refs. [25–30,39,51–59] and from ref. [60] for MORB and ref. [61] for the continental crust composition.

**Construction of the melt model**. A partial melting model was constructed to evaluate the nature of the sources of rejuvenated lavas and nakhlites and chassignites vs. the sources of Hawaiian main shield-stage lavas and shergottites. Neither depleted (olivine (Ol), orthopyroxene (Opx), spinel (Sp), or garnet (Gt)) nor fertile mantle compositions (olivine, orthopyroxene, clinopyroxene (cpx), spinel, or garnet) can produce the trends in Fig. 5, using the partition coefficients summarized in https://earthref.org/KDD/ and in refs. [62–64] (Ol/Opx/Cpx/Gnt/ Spinel/Amp/Phlog/Rutile/Zircon = [Nb] 0.0005/0.0013/0.0067/0.001/0.8/0.2/ 0.088/50/0; [Zr] 0.0031/0.013/0.11/0.14/0.6/0.2/0.17/4/100; [Y] 0.0057/0.074/0.4/3/

0.05/1.4/0.018/0.76/0.7). Using a primitive mantle composition[65] and a fertile mantle source assemblage (60% Ol, 17–20% Opx, 15% Cpx, 5–8% Gt or Sp) with modal melting proportions of phases can reproduce the Hawaiian shield-stage lavas between 3 and10% fractional partial melting. At low degrees of partial melting (~0.1%), such a source would also lead to a composition like Hawaiian rejuvenated lavas, but it would not be able to reproduce the Sr-Nd isotope compositions that required a long-term depleted mantle source.

We can reproduce the range of MORB compositions assuming a depleted MORB mantle composition[66] and a correspondingly depleted mineral source composition (70% Ol, 25% Opx, 5% Gt or Sp) and dominant melting of olivine and orthopyroxene after exhaustion of garnet and spinel, with partial melts between 5 and 10% for the most depleted MORB samples. This trend can partly explain the martian shergottite compositions, although in detail, the depleted shergottites require an even more depleted source, evident from the highly unradiogenic Sr and radiogenic Nd compositions of these samples.

For shergottites, we can generate the range of Nb/Y and Zr/Y observed in the depleted, intermediate, and enriched shergottites by exhausting garnet (depleted shergottites) and retaining progressive amounts of garnet in the residue (intermediate and enriched shergottites), consistent with prior studies[14].

Despite the ability of a primitive mantle composition partial melt model to generate melts similar to Hawaiian rejuvenated lavas, such a source is implausible given the unradiogenic Sr and radiogenic Nd isotope signatures of these lavas, suggesting a depleted source. To reproduce compositions similar to nakhlites and chassignites and rejuvenated Hawaiian lavas, a source with a mixture of phases that preferentially fractionate Zr—and in particular Nb—from Y are required. Phases with these characteristics include hydrous phases (amphibole, phlogopite) and rutile or ilmenite. Rutile has high affinity for Nb and, along with ilmenite, amphibole, and mica, have been recognized in terrestrial metasomatized lithosphere (MARID—mica–amphibole–rutile–ilmenite–diopside[67]). To date, there is no evidence for such occurrences in the martian mantle, but the presence of amphibole and micas in some martian meteorites suggests that hydrated mineral associations are possible in the martian mantle[22]. Modeled melting of a depleted mantle composition with ~1.5ppm Y and Nb/Zr = 0.01 with 0.05% rutile and 1% amphibole will result in rapid and complete exhaustion of exotic phases, and high resultant Nb/Y for a given Zr/Y value at between 0.1 and 1% partial melting of the martian-depleted mantle or the terrestrial-depleted mantle.

**High field strength elements and barium**. The elements Ti, Zr, Y, and Nb are all high field strength elements (charge/radius ratio) that are not usually transported in aqueous fluids or strongly affected by metamorphic processes. These properties, and their systematic variation in lavas, make these elements important tracers of source variations in volcanic rocks[68,69]. In plots of these elements vs. MgO, there is clear distinction in the shergottite geochemical groups (enriched, intermediate, depleted) for La/Yb, Nb/Y, and Zr/Nb (Fig. 3). While Ba is similarly incompatible to Nb and La, it can be strongly modified by terrestrial alteration, most notably in depleted shergottites (e.g., DaG 476). The similarity in Ce/Pb could implicate melts with enriched shergottite-like characteristics as being similar to those that were lost from the source of nakhlites and chassignites.

**Crystallization ages**. Crystallization ages for shergottites were taken from the literature (Supplementary Data 1) and Rb-Sr and Sm-Nd isotope crystallization ages were favored to plot the Sr-Nd isotope diagram (Fig. 1). Nakhlites and chassignites have been dated between 1270 and 1420 Ma, with an estimated mean age of 1340 ± 40 Ma on the basis of 40 individual age determinations (see ref. [20], and data in refs. [40,41,45,47,49,70]).

## Data availability
The data that support the findings of this study are available within the paper and within the Supplementary Data file.

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

## Acknowledgements

Financial support to complete this work came from the NASA Solar System Workings program (NNX16AR95G). J.M.D.D. acknowledges funding from the ANR-Idex Sorbonne Paris Cité for a long-term visiting program. F.M. acknowledges funding from the European Research Council (ERC Starting grant agreement 637503-Pristine) and the UnivEarthS Labex program at Sorbonne Paris Cité (ANR-10-LABX-0023 and ANR-11-IDEX-0005-02). Y.L. contribution was conducted at the Jet Propulsion Laboratory, which is managed by the California Institute of Technology under contract with NASA. Y.L. also acknowledges the support from JPL-Caltech President's and Director's Fund. We are grateful to Margaret Deng and Allison Kubo for their assistance in the lab.

## Author contributions

J.M.D.D. designed the project and wrote the paper. J.M.D.D., K.T.T., A.U., F.M., and Y.L. obtained samples. J.M.D.D., K.T.T., and A.U. performed analyses. J.M.D.D., K.T.T., A.U., F.M., Y.L., and C.R.N. all participated in the interpretation of the data and provided input on the manuscript.

## Additional information

**Competing interests:** The authors declare no competing interests.

