## [Peer Review File · Nature Communications]

Reviewers' Comments:

Reviewer #1:

Remarks to the Author:

This paper by Day et al. reports the results of new trace element analyses from 25 shergottite meteorites. The paper also uses chemical data from two other manuscripts not yet published, specifically: major element data of 39 meteorites (shergottites, nakhlites, and chassignites), and trace element data for 16 nakhlites+chassignites. This number of analyses, all from the same laboratory, and using the same techniques provides a very comprehensive geochemical dataset, and will be highly valuable for martian research. My comments on the manuscript are outlined below.

The authors have demonstrated that the nakhlite+chassignite meteorites are consistent with having been produced from metasomatized mantle, different to the mantle melted to form the shergottite meteorites. This is an interesting and important result. However, saying that the nakhlite+chassignite magmas are 'rejuvenated stage' eruptions is rather more tenuous, and should be stated with more caution.

On Earth (e.g., Hawaii), shield stage and rejuvenated magmas have very specific and distinct temporal and spatial characteristics, with the rejuvenated magmas erupted a few hundred kilometers down the volcanic chain away from the main plume, and a few million years after the main shield stage (Clague, 1987). Martian meteorites, however, lack these spatial and temporal links. Being meteorites, we have little idea of their source localities on Mars. No source craters are yet confirmed (e.g., Herd et al., 2018), so the sources for the nakhlites/chassignites vs. shergottites could potentially be located on the opposite sides of the planet. The only constraints are from cosmogenic exposure ages, and these results indicate the shergottites and nakhlites/chassignites have different exposure ages, and are therefore sampling different craters, and confirm that these rocks are sampling different locations on the martian surface (Herzog and Caffee, 2014).

The available temporal data for the eruption ages of the shergottites (~2400 to 165 Ma) vs. nakhlites/chassignites (~1340 Ma) are also inappropriate for the nakhlites to be rejuvenated magmas. Day et al. state this in the manuscript (lines 177-178) "Additionally, shergottite crystallization ages (165-2400 Ma) preclude a direct link between nakhlites and younger shergottites." Why put the samples into a very narrow and specific box ('rejuvenated magmas') that is not supported by the available data (as stated on lines 177-178), when the data works very well in a more general – but still highly important box of 'melting metasomatized mantle'?

Furthermore, on Earth, there are several different ways to get melting of metasomatized mantle, for example (but not restricted to): edge-driven convection (Demidjuk et al., 2007), lithospheric flexure (Pilet et al., 2016), or metasomatism of recycled continental lithosphere (Hoernle et al., 2011). Clearly on Earth, magmatism via melting of metasomatized mantle can be produced in settings other than rejuvenated magmatism, and some of these mechanisms could also apply for Mars.

In summary, the term 'rejuvenated' requires spatial and temporal links that are not demonstrated by the shergottites and nakhlites/chassignites. Could the paper be instead titled and written as "Martian magmatism from plume metasomatized mantle"?

Other important points:

All diagrams should have uncertainty estimates plotted, or a statement that uncertainties are smaller than the data point symbols. The analytical uncertainty for each element has already been determined – either by using the long-term reproducibility of ~6% (line 206), or the %RSD values listed for each element in Table S1, as determined via replicate measurements of international standards BCR-2 and AGV-2. These values are non-trivial: the %RSD values listed in Table S1 are

>5% for many elements, e.g., REEs.

Methods (also lines 80-81). The use of identical analytical protocols does indeed remove non-systematic bias between samples, and as such this data will make a very important contribution to the study of martian meteorites. However, the authors need to comprehensively evaluate 'non-analytical' aspects of their data, especially 1) the effects of terrestrial weathering, and 2) sample heterogeneity and the 'nugget effect'. Most of the new text required could go in either main manuscript or in the "methods" section – which at the moment is very brief (lines 190-206).

In terms of terrestrial weathering, Day et al have briefly discussed the effects of hot desert weathering on selected trace elements Ba, Sr, and U (Lines 90-91). However, other trace elements or major elements are also likely to be influenced by hot or cold desert weathering – what about the concentrations of (for example, but not limited to) Ca, K, or Na? In certain circumstances, K and Na can be redistributed by water (Rollinson, 1993), so could be removed, while calcite is present in hot desert 'caliche' (Treiman, 2005), so concentrations of Ca may be higher than in the original igneous rock. The major elements are important because they are used for the geochemical modeling and also in Figure S1.

In terms of sample heterogeneity, the authors need to 1) state the total mass of rock that was crushed in order to produce the powders used for major and trace element analysis, and 2) evaluate if this mass was sufficient to ensure the powder is representative of that meteorite. At the moment, the only detail in this regard was that ~50 mg was used for trace-element analysis (line 197) – presumably a much larger mass was actually crushed, in order to produce a (reasonably) representative sample. An important point is that if too small a mass was crushed, then the analyses may suffer from the 'nugget effect' (e.g., <https://www.911metallurgist.com/blog/gold-nugget-effect-definition-in-sampling>). In the case of martian meteorites, the 'nuggets' would be represented by rare phases like apatite or zircon, which are not uniformly distributed throughout the meteorite, but instead concentrated in the last melt to crystallize (e.g., Treiman, 2005). Rare earth elements (for example) are highly concentrated in trace phases like apatite – if insufficient material was crushed, then these 'nuggets' may have been missed, and the chemical analysis would not be representative of the rock.

Day et al have replicate analyses of several meteorites (Table S1). The reproducibility of these analyses needs to be discussed – using the REE as an example, there is considerable variation between analyses for numerous elements in some meteorites: for example, > 15% for NWA 3137, >20% for Tissint, and often >20% for Nakhla. These discrepancies need to be addressed, as the differences are much greater than the %RSD values obtained for the rock standards BCR-2 and AGV-2, or the long term reproducibility of ~6% (line 206). Could these greater discrepancies observed for the duplicate analyses of the martian samples be due to the nugget effect?

Fortunately, however, the chemical concentrations observed in the shergottites vs. nakhlites/chassignites can vary by as much as an order of magnitude (e.g., Figure 2), so the differences should exceed these analytical effects, and should not influence the major conclusion of this study.

Ages of the meteorites. For the calculation of isotopic values, the authors have this statement (lines 345-346) "Age-corrected Sr and Nd isotope data for shergottites, nakhlites and chassignites in Figure 1 were obtained from Refs. 2, 30-41, and references therein." The text on lines 345-346 is ok, however, the authors also have the following on lines 405-407: "Crystallization ages for shergottites were taken from the literature and Rb-Sr and Sm-Nd crystallization ages were favored to plots the Sr-Nd diagram (Figure 1)." This statement of "taken from the literature" is too vague to be reproducible by subsequent researchers. Either the text "taken from the literature" should be replaced with text similar to that already on lines 345-346, or (ideally) the exact age and uncertainty used for each meteorite be listed somewhere in the manuscript, along with a reference

for that age. A possibility would be to add this information in Table S1.

The age of the nakhlites and chassignites (lines 406-407) "has been estimated at 1340 ± 40 Ma based on 40 individual ages (Ref. 61)." Reference 61 is unfortunately not yet published, so this statement is difficult to evaluate in this review. However, I will make two comments:

- 1) The age stated for the nakhlites/chassignites is not consistently reported. In the supplementary material it is " 1340 ± 40 Ma" (line 407), while in the abstract it is stated at " ~ 1340 Ma" (line 17), while in other locations in the main text the age is given as an exact " 1.34 Ga" (e.g., line 67, line 177). The age reported should be consistent.
- 2) It is becoming increasingly apparent (including material published by authors of the current manuscript, e.g., Udry and Day, 2018), that the nakhlites and chassignites represent multiple eruption events. If the nakhlites/chassignites have multiple eruptions, and consequently different ages, then why not report the full range of ages represented (i.e., like the range of ~ 165 - 2400 Ma reported for the shergottites), rather than just an 'average' age?

As a more general point, as Nature Communications is read by a wide range of scientists, not just martian meteorite experts and geologists, I would suggest that the authors consistently use "Ma" for ages, rather than swapping between Ga and Ma. Being consistent with the units will also facilitate age comparisons between the different groups.

Specific comments:

- The abstract is far too long for Nature Communications.
- The abstract should not contain references.
- Line 46-47: "For example, observed pre-shield, shield and post-shield erosional and rejuvenated stages of volcanism observed in Hawaii have not been recognized for Mars" This statement is true – but as the martian lithosphere is stationary, and not moving over martian plumes, should Mars necessarily be expected to have the same behavior as on Earth?
- Line 51: the composition of martian atmosphere was also measured by Curiosity (Mahaffy et al., 2013). As the values from Curiosity are more precise, they should also be mentioned here.
- Line 52 (also lines 61, 100, 128, 135, 138, 376, 385, and potentially elsewhere) references should be formatted as a superscript.
- Line 54: NWA 7034 and NWA 7533 are not the only meteorites in this pairing group (see <http://www.imca.cc/mars/martian-meteorites-list.htm>), so I would suggest this text is changed to "NWA 7034/7533 and pairs".
- Line 60: "Increasingly, this relationship has been recognized as being inconsistent with available petrological and geochemical data...". The science behind this statement seems correct, but I think the statement would benefit from a brief expansion to say why the petrologic and geochemical data is inconsistent with the idea of depleted reservoir = mantle and enriched reservoir = crust.
- Line 69: the nakhlite meteorites should not all be described as "clinopyroxenites". In the IUGS classification scheme for igneous rocks, clinopyroxenite is a term that is restricted to ultramafic igneous rocks where all crystals are visible in thin section. In contrast, most meteorites in the nakhlite group have chemical analyses of mafic composition and significant (~ 8 - 35 vol. %) fine-grained mesostasis where the percentages of all individual minerals cannot be determined (Treiman, 2005; Tomkinson et al., 2015). The IUGS states that in such cases where the mineral mode cannot be determined, then the chemical classifications related to the total-alkalis silica (TAS) diagram should be used (Le Maitre, 2002). As is clearly demonstrated by Figure S1 of this manuscript, most of the nakhlites are basaltic in composition, with mafic levels of between 45-52 wt % SiO₂, although there are some meteorites that plot in other fields (basaltic andesite, mugearite, foidite) – as is noted by Day et al. on line 84. The nakhlites should therefore be described in a more general (and more accurate) way, for example "pyroxene-rich mafic rocks". The literature on nakhlites has a long history of using the term 'clinopyroxenites' – but this incorrect name according to IUGS procedures should not be perpetuated into the future – especially for a paper that provides clear data (Table S1, Figure S1) showing why such a term is not appropriate. This is quite a lot of text for a single word, but accurate classification is a crucial part of science, as names greatly influence perceptions.

- Line 78: the new analyses reported in this paper are for shergottite trace element analyses only; in the footnote to Table S1, the authors state that the major element data is from other sources, as is the data for the nakhlites and chassignites. This statement on the data source should also appear in the main manuscript or methods section – it's presently too difficult to find as only a footnote to Table S1. Possibly "We present new trace-element data for 25 shergottite meteorites and compare these data with trace element analyses obtained for nakhlites/chassignites using identical analytical protocols (Ref X). Major element analyses are by method Z, with additional analytical details reported in Refs X and Y."
- Line 78 (and elsewhere): states that "25 shergottites" were analyzed. However, in Table S1, I could only count new trace-element analyses for 23 shergottites. Could the authors please confirm the number of new shergottite analyses.
- Line 87 and many other places in the manuscript. The "±" symbol should be consistently formatted with a space on either side of the symbol, i.e., "number space ± space uncertainty" (0.114 ± 0.044).
- Line 87: please specify the statistical measurement being reported by the ± number. i.e., is this 1 sigma or 2 sigma? Standard deviation or standard error of the mean?
- Line 92: "Sample NWA 6963 is exceptional in that it has the major element geochemical composition of a shergottite, but the incompatible trace element composition of a nakhlite." For this meteorite please briefly elaborate or show on a diagram which major elements are like the shergottites, and not like nakhlites. As far as I could see, NWA 6963 falls within the nakhlite range for all elements except Al and Fe, where NWA 6963 has more Al but less Fe than the nakhlites reported in this study. However, Day et al did not analyze NWA 5790, which is currently the nakhlite with the most mesostasis (Le Maitre, 2002). As the mesostasis is rich in feldspar [i.e., high Al, low Fe]), chemical analysis of NWA 5790 may expand the range of major elements for the nakhlites, potentially removing the difference between NWA 6963 and the nakhlites.
- Lines 120-122: "Such characteristics cannot be explained by crustal contamination of parental magmas, since both terrestrial and martian crust (represented by NWA 7034) have lower Nb/Y than nakhlites/chassignites or Hawaiian rejuvenated lavas." Saying "cannot" is a very definitive statement, and may be over-reaching the interpretations possible with the available suite of samples. The meteorite NWA 7034 and pairs only sample one locality of the martian regolith; areas of the martian crust and lithospheric mantle (i.e, existing at depths beyond the deposition of impact breccia debris) will very likely be far more heterogeneous – think of the heterogeneity of different lithospheric terrains on earth, both aerially and at depth.
- Lines 120-122: this sentence is also poorly worded – as written, NWA 7034 represents both terrestrial and martian crust. I would suggest changing the order to "since both martian regolith (represented by NWA 7034 and pairs) and terrestrial crust have lower Nb/Y..."
- Line 134: Tharsis is not the only region of massive magmatism on Mars. I would suggest rewording to "In detail, the region with the most massive volcanism – Tharsis – ..." or "In detail, a region with massive volcanism – Tharsis – ...".
- Line 136: "young volcanism" means different things in different fields, so please define (in Ma) what is meant by this statement. An eruption at 600 Ma is 'young' for Mars – but on Earth, that's in the Precambrian...
- Lines 124-138: How does this text describe a mechanism to form rejuvenated magmas?
- Lines 165-166: "In detail, however, differences in isotopic compositions suggests that, while there has been plate tectonics on Earth, it has not occurred for Mars." I've a few issues with this statement. 1) Even without plate tectonics, convection can and does occur on rocky planets. 2) The martian mantle is clearly still convecting (e.g., in order to produce late Amazonian volcanoes like Tharsis). 3) Convection will act to homogenize isotopic reservoirs, so it is more accurate to state that the preservation of distinct geochemical reservoirs indicates that convection (not plate tectonics) has been unable or less able to homogenize these reservoirs on Mars. 4) Even with mantle convection, the preservation of reservoirs only indicates that some parts of Mars are able to preserve geochemical differences. One possible candidate for a resistant reservoir (on Mars, as it is on Earth) is the thick lithospheric mantle keel – which is even more pronounced on Mars than on Earth. In this manuscript do the authors want to consider the geochemical influence of a thick lithosphere? 5) I would recommend stating something like "convection on Mars was much less

efficient in homogenizing isotopic reservoirs”.

- Lines 184-185. To me the sentences on these lines are a bit awkwardly placed, and there does not seem to be a good link between them.
- Lines 185-186. “Nakhlites/chassignites represent melting of metasomatized mantle from early magmatic events. These meteorites could have been ejected from deep craters, or are from regions distinct from younger shergottites.” I think there are a few issues with this statement about deep craters. The logic in part of this sentence seems to be: nakhlites+chassignites are earlier than (most of) the shergottites, therefore the nakhlites+chassignites would be buried beneath the shergottites, and therefore must be from deeper in the craters. This is unnecessarily complicated, and actually goes against the evidence from other studies of martian meteorites. Firstly, we already know that the nakhlites+chassignites are from different craters to the shergottites, as these different meteorite groups have different cosmogenic exposure ages (Tomkinson et al., 2015). Secondly, we know from NASA mapping that ~1.4 Ga volcanic rocks are widely exposed at the surface of Mars (Herzog and Caffee, 2014), so there is no need to hypothesize that the nakhlites must be ejected from deep within craters.
- Various figures: The chassignites and nakhlites are not identical, so it would be beneficial if these rock types had different symbols.
- Figure 3: not all of the lines are described in the figure caption.
- Figure S1: Samples that fall in the “foidite” can be classified further (Le Maitre, 2002).
- Table S1: What is the “TAB” data in the last three columns? Make sure that all abbreviations or column headings are adequately described.

In summary, the manuscript contains an important suite of analyses that will be very useful for examining martian meteorites, martian volcanism, and the geochemical evolution of that planet. However, more work needs to be done in the reporting of the methods, evaluation of the analytical results, and examination of the scientific ideas. At the moment the manuscript seems very short for Nature Communications. Therefore there is ample space for expansion of the text and number of figures, which will allow for considerable enhancement to the scope of the manuscript.

References:

- Clague, D.A., 1987. Hawaiian alkaline volcanism. In: Fitton, J.G., Upton, B.G. (Eds.), *Alkaline Igneous Rocks*. Blackwell Scientific Publications, Oxford, pp. 227-252.
- Demidjuk, Z., Turner, S., Sandiford, M., George, R., Foden, J., Etheridge, M., 2007. U-series isotope and geodynamic constraints on mantle melting processes beneath the Newer Volcanic Province in South Australia. *Earth and Planetary Science Letters* 261, 517-533.
- Herd, C.D.K., Tornabene, L.L., Bowling, T.J., Walton, E.L., Sharp, T.G., Melosh, H.J., Hamilton, J.S., Viviano, C.E., Ehlmann, B.L., 2018. Linking martian meteorites to their source craters: new insights. *LPSC*, p. Abstract #2266.
- Herzog, G.F., Caffee, M.W., 2014. Cosmic-Ray Exposure Ages of Meteorites. In: Davis, A.M. (Ed.), *Treatise on Geochemistry 2nd Edition*. Elsevier, Amsterdam, pp. 419-454.
- Hoernle, K., Hauff, F., Werner, R., van den Bogaard, P., Gibbons, A.D., Conrad, S., Müller, R.D., 2011. Origin of Indian Ocean Seamount Province by shallow recycling of continental lithosphere. *Nature Geoscience* 4, 883-887. doi:10.1038/NNGEO1331
- Le Maitre, R.W., 2002. *Igneous Rocks: A Classification and Glossary of Terms*. Second ed. Cambridge University Press, Cambridge.
- Mahaffy, P.R., Webster, C.R., Atreya, S.K., Franz, H., Wong, M., Conrad, P.G., Harpold, D., Jones, J.J., Leshin, L.A., Manning, H., Owen, T., Pepin, R.O., Squires, S., Trainer, M., MSL Science Team, 2013. Abundance and isotopic composition of gases in the Martian atmosphere from the Curiosity Rover. *Science* 341, 263-266. doi:10.1126/science.1237966
- Pilet, S., Abe, N., Rochat, L., Kaczmarek, M.A., Hirano, N., Machida, S., Buchs, D.M., Baumgartner, P.O., Müntener, O., 2016. Pre-subduction metasomatic enrichment of the oceanic lithosphere induced by plate flexure. *Nature Geoscience* 9, 898-903. doi:10.1038/ngeo2825
- Rollinson, H., 1993. *Using Geochemical Data: evaluation, presentation, interpretation*. Prentice Hall, London.
- Tomkinson, T., Lee, M.R., Mark, D.F., Dobson, K.J., Franchi, I.A., 2015. The Northwest Africa

(NWA) 5790 meteorite: A mesostasis-rich nakhlite with little or no Martian aqueous alteration. *Meteoritics & Planetary Science* 50, 287-304. doi:10.1111/maps.12424
Treiman, A.H., 2005. The nakhlite meteorites: augite-rich igneous rocks from Mars. *Chemie der Erde - Geochemistry* 65, 203-270. doi:10.1016/j.chemer.2005.01.004
Udry, A., Day, J.M.D., 2018. Formation and emplacement of the cogenetic nakhlite and chassignite meteorites. LPSC, p. Abstract #1052.

Reviewer #2:

Remarks to the Author:

See attached file.

Reviewer #3:

Remarks to the Author:

Review of

"Rejuvenated martian magmatism from plume metasomatized mantle"

by JMD Day et al.

It is challenging to reconstruct the major petrogenetic processes occurring on a planet with only a limited and probably biased set of samples. This is definitely the case with Mars, and authors of the submitted manuscript rightly claim that no satisfactory unified model has been put forth to explain the petrogenesis of shergottites, nakhlites, and chassignites, the SNC suite of meteorites. Using a new, internally consistent dataset of trace element concentrations from 25 shergottites, together with existing nakhlite data and a simple melt model, the authors draw on analogies between shergottites and the Hawaiian main shield basalts and between nakhlites (and chassignites) and the products of Hawaiian "rejuvenated" volcanism to explain the petrogenetic relationship among SNC members. This is a novel and very interesting concept and it makes predictions about the spatial relationships between these rock types that could be tested by future missions to volcanic edifices on Mars. Particularly thought-provoking is the focus on plume magmatism, to draw petrogenetic parallels between a tectonically active Earth on the one hand and Mars, with its lack of plate tectonics on the other.

The authors focus mainly on only three of the suite of 41 analyzed trace elements, i.e., Nb, Zr, and Y. These have the advantage of being sensitive to certain igneous differentiation processes, while being fluid immobile and thus expected to be intact in meteorites known to have undergone hot- or cold desert weathering and alteration. On the basis of higher Nb/Y at a given Zr/Y of nakhlites and rejuvenated Hawaiian lavas, and the fact that the later have a contribution from a metasomatized depleted source, the authors suggest (reasonably) that a similarly metasomatized source can explain the nakhlite (and chassignite) compositions.

Despite the novel ideas presented here, I have some misgivings outlined in my main comments below that prevent me from recommending this for immediate publication. I think these would be easily addressed in a larger, more detailed article submitted elsewhere.

* Ruling out effects of surface processes on Earth and Mars:

Many martian meteorites have been affected by terrestrial weathering, alteration, and contamination (the latter meaning addition of fine-grained material during terrestrial residence or sawing). These will probably have little effect on Nb, Zr, and Y, but it would be good to demonstrate this. Sawing off fusion crust is a good start, but how can the other effects be monitored and determined to be insignificant? When adding isotope ratios and other trace element constraints to the model (see below), significant effects of surface processes need to be confidently ruled out first.

* Analytical methods: The digestions were done at low pressure on a hotplate. Therefore some refractory HFSE-bearing phases might not have been completely digested. It needs to be shown (perhaps using other digestion methods that attack HFSE minerals) that this has not affected the axes of the main Fig 3 plot. To their credit, the authors have supplied a sufficient number of geologic standard analyses to allow proper assessment of data quality.

* The partial melting model needs to be much better documented and explained than it is. This means including 1) a table of the actual partition coefficients used (not just a catch-all reference to a large database plus 3 specific papers, 2) the calculations used to define model gridlines and melting arrows on Fig. 3, and 3) a realistic assessment of error propagations. It is difficult to see the primitive and depleted mantle models under the data points on Fig. 3. The difference between dashed lines and solid lines also needs to be explained.

* The presented novel hypothesis of the Hawaiian rejuvenated volcanism analog departs significantly from earlier petrogenetic models (and this a good thing). A successful hypothesis explains all existing data, not just a subset of trace elements. Although the authors qualitatively discuss major element and isotopic constraints in the context of their model, their idea would be more convincing if backed up by quantitative constraints from existing isotope data, as well as other trace- and major element data. The authors might also want to consider oxygen fugacity constraints (e.g., Herd, C.D.K., 2003. The oxygen fugacity of olivine-phyric martian basalts and the components within the mantle and crust of Mars. *Meteoritics & Planetary Science* 38, 1793–1805.) The authors should also think about how far removed individual meteorites are from parental melt compositions (via fractional crystallization, crystal accumulation, or assimilation). This issue was highlighted early on: Longhi, J., 1991. Complex magmatic processes on Mars- Inferences from the SNC meteorites. *Proc. Lunar Planet Sci.* 21, 695–709. Also note the

Specific comments:

24: what does "lower coupled Nb/Y-Zr/Y" mean? Lower Nb/T at a given Zr/Y.

68-69: "depleted Rb/Sr and Sm/Nd isotopic compositions" – I know what is meant, but Rb/Sr and Sm/Nd are primarily element ratios, not isotopic compositions. I recommend "depleted Sr and Nd isotopic compositions"

72: Again, be careful with the use of "enriched" and "depleted" with element ratios. Here "Sm/Nd enriched" does not make sense. "Enriched" means incompatible element enriched, i.e., a lower Sm/Nd compared to the reference reservoir. To explain ¹⁴²Nd enrichments however, the Sm/Nd had to be higher than the reference reservoir.

Adjective-noun & subject-verb agreement:

21: "dataset...reveals"

157: "basalt compositions"

165: "differences...suggest"

178: "Nakhlites come"

382: " a source...is required"

406: "plot the diagram"

Style suggestions:

Avoid "and/or." Logically, "or" includes the option of "and" and is thus not equivalent to "either...or" (Line 52)

Avoid ambiguous or non-committal slash "/" constructions. Does the slash denote a ratio, alternative choices, "or," or "and"? Line 140 gives a particularly ambiguous example: "shield stage or pre-/post-shield stage" covers all possibilities.

Replace "nakhlites/chassignites" with "nakhlites and chassignites" at first and perhaps with "NC"

afterwards for brevity.

Avoid anthropomorphisms. Rocks do not "experience" and have not "seen" anything. (Lines 42, 179)

FIGURE 2: Cite data sources clearly; Note that none of the Nakhlite data are from the present study.

FIGURE 3:

With the exception of the metasomatised source model, the models in the figure are difficult to see or follow.

What is the significance of the grey dotted lines?

339: "martial"->"martian"

SUPPLEMENTARY MATERIALS:

355, 356: Both assemblages should be clarified by adding the word "and" before "spinel or garnet"

FIGURE S1: The caption should more directly label the data sources. According to the caption of Table S1, no major element data are from the current study.

TABLE S1:

According to the table caption, the major element data for the shergottites, and the major- and trace element data for nakhlites are being primarily reported in two papers submitted to EPSL. It seems inappropriate to include the data in the current data table until those papers have been accepted.

Most major element concentrations are displayed at a reasonable number of significant digits (which happens to correspond to two decimal places for the major elements). However, no trace element data measured by concentration reference ICP-MS should be displayed to more than 3 significant digits. It is especially not warranted when the external reproducibility is percent-level or worse ("better than 6%" for most elements; >10% for Mo, Te, and Se.) It would also be good to round off the values in the table to the correct number of significant digits rather than to maintain 15 digits of precision in each cell.

For facilitating future references to this data set, it would be good to provide each analysis with a unique ID number or at least label replicates a, b, c, etc.

What do the "TAB" columns at the end mean?

Some SiO₂ contents (NWA 7721, 6342) were measured by difference assuming 100% total. Why?

Review of Day et al. (Rejuvenated martian magmatism...)

By Jeff Taylor, U. Hawaii

This paper presents a coherent and extensive set of data on Martian meteorites and uses it to present an interesting, unified view of the petrogenesis of shergottites and nakhlites/chassignites. The Martian crust is decorated by prominent, large, basaltic volcanoes, whose origin is almost always ascribed to long-lived plumes from the mantle. The paper makes a good case for shergottite magmas being analogous to the shield-build magma that produce Hawaiian volcanoes and that the nakhlites are analogous to the rejuvenated stage. The important contribution of the paper (besides the excellent data presented in the supplement) and the reason it should be published is that it draws a coherent picture of Martian magma generation in the context of mantle plumes. Thus, I recommend publication. However, I have a few suggestions that could make the paper a bit more well-rounded, though at the risk of placing less emphasis on the integrated plume idea.

Meteorites may not be representative of the Martian crust: Shergottites and nakhlites are important sources of information about Mars, but they may not be abundant rocks. Mars Odyssey Gamma-Ray Spectrometry (GRS) shows that the crust is richer in incompatible elements (specifically K and Th) than the Martian meteorites (see Fig. 4 in Taylor et al., 2006), though they have about the same K/Th. Other parameters are similar, such as FeO (see Fig. 7 in Taylor et al., 2006). In addition, the igneous rocks at Gusev crater are richer in alkalis than are the shergottites (see McSween et al., 2006), although we do not have the complete set of elements that appears in Day et al. Gusev does house some olivine basalts, too. Perhaps it would be more complete to point out the differences in bulk crustal compositions from GRS and the nature of Gusev (and Gale, too) rocks compared to Martian meteorites. What could be the relation of the bulk crust and the Gusev rocks to the plume idea?

NWA 7034 etc. seems very different from the shergottites. The Black Beauty breccia opens up other questions. It has alkalic, gabbroic clasts in it. Do these relate to one of the plume components, or do they represent another form of magma production in Mars? U-Pb and Sm-Nd dating indicates that they are very old, 4.42 Ga, so one might argue that they formed by a mechanism different from plumes, if the plume phase of Martian mantle evolution started later. I have no answers, but it could be useful to mention these rocks. Some of the key elements to test the fit to the plume idea have been analyzed and presented in the Humayan et al. paper on NWA 7034.

No volcano in Gusev. To return to the Gusev rocks again, if the olivine basalts and alkalic basalts indicate a main shield stage and a rejuvenation stage indicate a plume-related process, where is the volcano?

Ferrobasalts on Earth. Justin Filiberto (2008) wrote an interesting paper discussing terrestrial analogs for the shergottites. He drew attention to ferrobasaltic rocks in the oldest eruptives in flood basalt provinces and as effusive flows and intrusive sills in some ocean floor locales such as Kola Peninsula. An important point is that these flows are associated with voluminous tholeiitic flows, hence perhaps making the connection between rejuvenation and shield-building stages. In addition, ideas for their petrogenesis include plumes and creation of a metasomatized mantle region (summary in section 3.1 of Justin's paper). These rocks might enhance the argument for a plume-style model. Or might show it is not always needed to explain a suite of rocks.

Heterogeneous mantle produced during magma ocean crystallization, and mantle sources in general.

A conventional idea is that the two shergottite sources, enriched and depleted, formed early in the history of Mars, by processes in the magma ocean. Does the plume model rule this out? Does it apply to initial creation of mantle sources? One can imagine a heterogeneous mantle still having plumes that behave something like described in the Day et al. manuscript. Also, others have drawn attention to diversity of mantle sources, including some compositional features that require missing end members. For example, see Fig. 6c in Taylor et al. (2008, in Jim Bell's Big Book), shows that all Martian meteorites are on or above the terrestrial (= chondritic) line on a plot of Ba/La vs La. Does this have anything to do with the plume idea? Does it weaken the hypothesis? (I doubt it, but it is worth pointing out this complexity. (Black Beauty falls on the terrestrial line.)

Filiberto, J. (2008) Similarities between the shergottites and terrestrial ferropicrites. *Icarus*, 197, 52-59.

Humayun, M., Nemchin, A., Zanda, B., Hewins, R. H., Grange, M., Kennedy, A., Lorand, J.-P., Göpel, C., Fieni, C., Pont, S., and Deldicque, D. (2013) Origin and age of the earliest Martian crust from meteorite NWA 7533. *Nature*, v. 503, p. 513-516. doi:10.1038/nature12764.

McSween et al. (2006) Alkaline volcanic rocks from the Columbia Hills, Gusev crater, Mars. . *Geophys. Res. 111*, doi:1029/2006JE002698.

Taylor, G. J., et al (2007), Bulk Composition and Early Differentiation of Mars, *J. Geophys. Res. 111*, E03S10, doi:10.1029/2005JE002645.

Taylor, G. J., McLennan, S.M., McSween, H. Y., Jr., Wyatt, M. B., and Lentz, R. C. F. (2008) Implications of observed primary lithologies. In *The Martian Surface: Composition, Mineralogy, and Physical Properties* (J. F. Bell, ed.), 501-513. Cambridge University Press.

Comments of Reviewer 1, Anonymous

This paper by Day et al. reports the results of new trace element analyses from 25 shergottite meteorites. The paper also uses chemical data from two other manuscripts not yet published, specifically: major element data of 39 meteorites (shergottites, nakhlites, and chassignites), and trace element data for 16 nakhlites +chassignites. This number of analyses, all from the same laboratory, and using the same techniques provides a very comprehensive geochemical dataset, and will be highly valuable for martian research. My comments on the manuscript are outlined below.

We thank the reviewer for their efforts in reviewing this manuscript.

The authors have demonstrated that the nakhlite+chassignite meteorites are consistent with having been produced from metasomatized mantle, different to the mantle melted to form the shergottite meteorites. This is an interesting and important result. However, saying that the nakhlite+chassignite magmas are 'rejuvenated stage' eruptions is rather more tenuous, and should be stated with more caution. On Earth (e.g., Hawaii), shield stage and rejuvenated magmas have very specific and distinct temporal and spatial characteristics, with the rejuvenated magmas erupted a few hundred kilometers down the volcanic chain away from the main plume, and a few million years after the main shield stage (Clague, 1987). Martian meteorites, however, lack these spatial and temporal links. Being meteorites, we have little idea of their source localities on Mars. No source craters are yet confirmed (e.g., Herd et al., 2018), so the sources for the nakhlites/chassignites vs. shergottites could potentially be located on the opposite sides of the planet. The only constraints are from cosmogenic exposure ages, and these results indicate the shergottites and nakhlites/chassignites have different exposure ages, and are therefore sampling different craters, and confirm that these rocks are sampling different locations on the martian surface (Herzog and Caffee, 2014).

While we agree that caution should always be applied to studying meteorites that have limited source information, our model by its virtue is a predictive tool. We have made every effort to add caution to the interpretation as classic 'rejuvenated' lavas on Earth, but the similarities are uncanny. Throughout the manuscript, we add some caution while also better explaining our model. We agree that exposure ages are useful, but in themselves are not diagnostic of source.

The available temporal data for the eruption ages of the shergottites (~2400 to 165 Ma) vs. nakhlites/chassignites (~1340 Ma) are also inappropriate for the nakhlites to be rejuvenated magmas. Day et al. state this in the manuscript (lines 177-178) “Additionally, shergottite crystallization ages (165-2400 Ma) preclude a direct link between nakhlites and younger shergottites.” Why put the samples into a very narrow and specific box (‘rejuvenated magmas’) that is not supported by the available data (as stated on lines 177-178), when the data works very well in a more general – but still highly important box of ‘melting metasomatised mantle’?

We agree with this statement, and we realize our intent was easily misread in the earlier version of the paper submitted to *Nature*. We point out the unique and uncanny similarity with Hawaiian rejuvenated lavas, and the fact that this mechanism is relevant on Mars. Modifications are made throughout the manuscript.

Furthermore, on Earth, there are several different ways to get melting of metasomatised mantle, for example (but not restricted to): edge-driven convection (Demidjuk et al., 2007), lithospheric flexure (Pilet et al., 2016), or metasomatism of recycled continental lithosphere (Hoernle et al., 2011). Clearly on Earth, magmatism via melting of metasomatised mantle can be produced in settings other than rejuvenated magmatism, and some of these mechanisms could also apply for Mars.

A useful point, but we don't see that these are clear alternative models. First, there is no evidence for edge-driven convection in Mars. Second, lithospheric flexure really isn't all that different from the rejuvenated Hawaiian lava model. Third, there is no evidence that metasomatism of recycled continental lithosphere could occur in Mars. Finally, many (exception is Demidjuk) of these texts lack the appropriate data with which to test if such models can generate the Nb/Y and Zr/Y variations observed. We cannot see how reference to these works would add to the discussion, so we have elected not to include them.

In summary, the term ‘rejuvenated’ requires spatial and temporal links that are not demonstrated by the shergottites and nakhlites/chassignites. Could the paper be instead titled and written as “Martian magmatism from plume metasomatised mantle”?

We agree. Our intention was misinterpreted and we have modified the title appropriately.

Other important points:

All diagrams should have uncertainty estimates plotted, or a statement that uncertainties are smaller than the data point symbols. The analytical uncertainty for each element has already been determined – either by using the long-term reproducibility of ~6% (line 206), or the %RSD values listed for each element in Table S1, as determined via replicate measurements of international standards BCR-2 and AGV-2. These values are non-trivial: the %RSD values listed in Table S1 are >5% for many elements, e.g., REEs.

Agreed. For major and trace elements, %RSD values better than 5% are quite hard to achieve consistently over the long-term life of a lab. We provide conservative and realistic errors; we doubt that long term reproducibility of better than 5% can be ‘reproduced’, especially for martian meteorites. This is therefore a state-of-the-art dataset.

Methods (also lines 80-81). The use of identical analytical protocols does indeed remove non-systematic bias between samples, and as such this data will make a very important contribution to the study of martian meteorites. However, the authors need to comprehensively evaluate 'non-analytical' aspects of their data, especially 1) the effects of terrestrial weathering, and 2) sample heterogeneity and the 'nugget effect'. Most of the new text required could go in either main manuscript or in the "methods" section – which at the moment is very brief (lines 190-206).

Agreed – we add more to the methods to address this important issue.

In terms of terrestrial weathering, Day et al have briefly discussed the effects of hot desert weathering on selected trace elements Ba, Sr, and U (Lines 90-91). However, other trace elements or major elements are also likely to be influenced by hot or cold desert weathering – what about the concentrations of (for example, but not limited to) Ca, K, or Na? In certain circumstances, K and Na can be redistributed by water (Rollinson, 1993), so could be removed, while calcite is present in hot desert 'caliche' (Treiman, 2005), so concentrations of Ca may be higher than in the original igneous rock. The major elements are important because they are used for the geochemical modeling and also in Figure S1.

Agreed – the effect of terrestrial weathering is nowhere near as severe as this reviewer might be led to believe. Nonetheless, we show this in the methods.

In terms of sample heterogeneity, the authors need to 1) state the total mass of rock that was crushed in order to produce the powders used for major and trace element analysis, and 2) evaluate if this mass was sufficient to ensure the powder is representative of that meteorite. At the moment, the only detail in this regard was that ~50 mg was used for trace-element analysis (line 197) – presumably a much larger mass was actually crushed, in order to produce a (reasonably) representative sample. An important point is that if too small a mass was crushed, then the analyses may suffer from the 'nugget effect' (e.g., <https://www.911metallurgist.com/blog/gold-nugget-effect-definition-in-sampling>). In the case of martian meteorites, the 'nuggets' would be represented by rare phases like apatite or zircon, which are not uniformly distributed throughout the meteorite, but instead concentrated in the last melt to crystallize (e.g., Treiman, 2005). Rare earth elements (for example) are highly concentrated in trace phases like apatite – if insufficient material was crushed, then these 'nuggets' may have been missed, and the chemical analysis would not be representative of the rock.

Day et al have replicate analyses of several meteorites (Table S1). The reproducibility of these analyses needs to be discussed – using the REE as an example, there is considerable variation between analyses for numerous elements in some meteorites: for example, > 15% for NWA 3137, >20% for Tissint, and often >20% for Nakhla. These discrepancies need to be addressed, as the differences are much greater than the %RSD values obtained for the rock standards BCR-2 and AGV-2, or the long term reproducibility of ~6% (line 206). Could these greater discrepancies observed for the duplicate analyses of the martian samples be due to the nugget effect?

Possibly. One of us (JD) routinely measures highly siderophile elements and has extensive knowledge of nugget effects. We have included a thorough analysis of this possible issue.

Fortunately, however, the chemical concentrations observed in the shergottites vs. nakhlites/chassignites can vary by as much as an order of magnitude (e.g., Figure 2), so the differences should exceed these analytical effects, and should not influence the major conclusion of this study.

Indeed.

Ages of the meteorites. For the calculation of isotopic values, the authors have this statement (lines 345-346) "Age-corrected Sr and Nd isotope data for shergottites, nakhlites and chassignites in Figure 1 were obtained from Refs. 2, 30-41, and references therein." The text on lines 345-346 is ok, however, the authors also have the following on lines 405-407: "Crystallization ages for shergottites were taken from the literature and Rb-Sr and Sm-Nd crystallization ages were favored to plots the Sr-Nd diagram (Figure 1)." This statement of "taken from the literature" is too vague to be reproducible by subsequent researchers. Either the text "taken from the literature" should be replaced with text similar to that already on lines 345-346, or (ideally) the exact age and uncertainty used for each meteorite be listed somewhere in the manuscript, along with a reference for that age. A possibility would be to add this information in Table S1.

Agreed. We have rectified this issue.

The age of the nakhlites and chassignites (lines 406-407) "has been estimated at 1340 ± 40 Ma based on 40 individual ages (Ref. 61)." Reference 61 is unfortunately not yet published, so this statement is difficult to evaluate in this review. However, I will make two comments: 1) The age stated for the nakhlites/chassignites is not consistently reported. In the supplementary material it is " 1340 ± 40 Ma" (line 407), while in the abstract it is stated at " ~ 1340 Ma" (line 17), while in other locations in the main text the age is given as an exact "1.34 Ga" (e.g., line 67, line 177). The age reported should be consistent. 2) It is becoming increasingly apparent (including material published by authors of the current manuscript, e.g., Udry and Day, 2018), that the nakhlites and chassignites represent multiple eruption events. If the nakhlites/chassignites have multiple eruptions, and consequently different ages, then why not report the full range of ages represented (i.e., like the range of ~ 165 -2400 Ma reported for the shergottites), rather than just an 'average' age?

The reviewer makes an important point. However, the issue raised of multiple flows from Ar-Ar age data for nakhlites alone is not so straight-forward. Independent evidence for multiple lava flows from petrological constraints (e.g., Udry & Day, in review) are not always correlated with new age data for nakhlites. The evidence for a range of ages related to multiple eruptions is not as straightforward as presented in Cohen et al. (2017), for example. We add a section outlining this issue.

As a more general point, as Nature Communications is read by a wide range of scientists, not just martian meteorite experts and geologists, I would suggest that the authors consistently use "Ma" for ages, rather than swapping between Ga and Ma. Being

consistent with the units will also facilitate age comparisons between the different groups.
Agreed, changed.

Specific comments:

The abstract is far too long for Nature Communications. The abstract should not contain references.

Fixed. This is a function of original submission to Nature.

- Line 46-47: “For example, observed pre-shield, shield and post-shield erosional and rejuvenated stages of volcanism observed in Hawaii have not been recognized for Mars” This statement is true – but as the martian lithosphere is stationary, and not moving over martian plumes, should Mars necessarily be expected to have the same behavior as on Earth?

No, we made this point, but now make it clearer.

- Line 51: the composition of martian atmosphere was also measured by Curiosity (Mahaffy et al., 2013). As the values from Curiosity are more precise, they should also be mentioned here.

Thank you. Added.

- Line 52 (also lines 61, 100, 128, 135, 138, 376, 385, and potentially elsewhere) references should be formatted as a superscript.

Fixed.

- Line 54: NWA 7034 and NWA 7533 are not the only meteorites in this pairing group (see <http://www.imca.cc/mars/martian-meteorites-list.htm>), so I would suggest this text is changed to “NWA 7034/7533 and pairs”.

Agreed, changed.

- Line 60: “Increasingly, this relationship has been recognized as being inconsistent with available petrological and geochemical data...”. The science behind this statement seems correct, but I think the statement would benefit from a brief expansion to say why the petrologic and geochemical data is inconsistent with the idea of depleted reservoir = mantle and enriched reservoir = crust.

Agreed, modified.

- Line 69: the nakhlite meteorites should not all be described as “clinopyroxenites”. In the IUGS classification scheme for igneous rocks, clinopyroxenite is a term that is restricted to ultramafic igneous rocks where all crystals are visible in thin section. In contrast, most meteorites in the nakhlite group have chemical analyses of mafic composition and significant (~8-35 vol. %) fine-grained mesostasis where the percentages of all individual minerals cannot be determined (Treiman, 2005; Tomkinson et al., 2015). The IUGS states that in such cases where the mineral mode cannot be determined, then the chemical classifications related to the total-alkalis silica (TAS) diagram should be used (Le Maitre, 2002). As is clearly demonstrated by Figure S1 of this manuscript, most of the nakhlites are basaltic in composition, with mafic levels of between 45-52 wt % SiO₂, although there

are some meteorites that plot in other fields (basaltic andesite, mugearite, foidite) – as is noted by Day et al. on line 84. The nakhlites should therefore be described in a more general (and more accurate) way, for example “pyroxene-rich mafic rocks”. The literature on nakhlites has a long history of using the term ‘clinopyroxenites’ – but this incorrect name according to IUGS procedures should not be perpetuated into the future – especially for a paper that provides clear data (Table S1, Figure S1) showing why such a term is not appropriate. This is quite a lot of text for a single word, but accurate classification is a crucial part of science, as names greatly influence perceptions.

Agreed, modified.

- Line 78: the new analyses reported in this paper are for shergottite trace element analyses only; in the footnote to Table S1, the authors state that the major element data is from other sources, as is the data for the nakhlites and chassignites. This statement on the data source should also appear in the main manuscript or methods section – it’s presently too difficult to find as only a footnote to Table S1. Possibly “We present new trace-element data for 25 shergottite meteorites and compare these data with trace element analyses obtained for nakhlites/chassignites using identical analytical protocols (Ref X). Major element analyses are by method Z, with additional analytical details reported in Refs X and Y.”

Good suggestion that we have followed – thank you

- Line 78 (and elsewhere): states that “25 shergottites” were analyzed. However, in Table S1, I could only count new trace-element analyses for 23 shergottites. Could the authors please confirm the number of new shergottite analyses.

Numbers are corrected appropriately.

- Line 87 and many other places in the manuscript. The “±” symbol should be consistently formatted with a space on either side of the symbol, i.e., “number space ± space uncertainty” (0.114 ± 0.044).

Changed.

- Line 87: please specify the statistical measurement being reported by the ± number. i.e., is this 1 sigma or 2 sigma? Standard deviation or standard error of the mean?

Modified.

- Line 92: “Sample NWA 6963 is exceptional in that it has the major element geochemical composition of a shergottite, but the incompatible trace element composition of a nakhlite.” For this meteorite please briefly elaborate or show on a diagram which major elements are like the shergottites, and not like nakhlites. As far as I could see, NWA 6963 falls within the nakhlite range for all elements except Al and Fe, where NWA 6963 has more Al but less Fe than the nakhlites reported in this study. However, Day et al did not analyze NWA 5790, which is currently the nakhlite with the most mesostasis (Le Maitre, 2002). As the mesostasis is rich in feldspar [i.e., high Al, low Fe]), chemical analysis of NWA 5790 may expand the range of major elements for the nakhlites, potentially removing the difference between NWA 6963 and the nakhlites.

Modified.

- Lines 120-122: “Such characteristics cannot be explained by crustal contamination of parental magmas, since both terrestrial and martian crust (represented by NWA 7034) have lower Nb/Y than nakhlites/chassignites or Hawaiian rejuvenated lavas.” Saying “cannot” is a very definitive statement, and may be over-reaching the interpretations possible with the available suite of samples. The meteorite NWA 7034 and pairs only sample one locality of the martian regolith; areas of the martian crust and lithospheric mantle (i.e, existing at depths beyond the deposition of impact breccia debris) will very likely be far more heterogeneous – think of the heterogeneity of different lithospheric terrains on earth, both aurally and at depth.

Agreed. Modified.

- Lines 120-122: this sentence is also poorly worded – as written, NWA 7034 represents both terrestrial and martian crust. I would suggest changing the order to “since both martian regolith (represented by NWA 7034 and pairs) and terrestrial crust have lower Nb/Y...”

Thanks for pointing this out. Modified.

- Line 134: Tharsis is not the only region of massive magmatism on Mars. I would suggest re-wording to “In detail, the region with the most massive volcanism – Tharsis – ...” or “In detail, a region with massive volcanism – Tharsis – ...”.

Modified.

- Line 136: “young volcanism” means different things in different fields, so please define (in Ma) what is meant by this statement. An eruption at 600 Ma is ‘young’ for Mars – but on Earth, that’s in the Precambrian...

A very good point! We have modified appropriately.

- Lines 124-138: How does this text describe a mechanism to form rejuvenated magmas?

Modified.

- Lines 165-166: “In detail, however, differences in isotopic compositions suggests that, while there has been plate tectonics on Earth, it has not occurred for Mars.” I’ve a few issues with this statement. 1) Even without plate tectonics, convection can and does occur on rocky planets. 2) The martian mantle is clearly still convecting (e.g., in order to produce late Amazonian volcanoes like Tharsis). 3) Convection will act to homogenize isotopic reservoirs, so it is more accurate to state that the preservation of distinct geochemical reservoirs indicates that convection (not plate tectonics) has been unable or less able to homogenize these reservoirs on Mars. 4) Even with mantle convection, the preservation of reservoirs only indicates that some parts of Mars are able to preserve geochemical differences. One possible candidate for a resistant reservoir (on Mars, as it is on Earth) is the thick lithospheric mantle keel – which is even more pronounced on Mars than on Earth. In this manuscript do the authors want to consider the geochemical influence of a thick lithosphere? 5) I would recommend stating something like “convection on Mars was much less efficient in homogenizing isotopic reservoirs”.

Agreed, our wording was loose. Modified.

- Lines 184-185. To me the sentences on these lines are a bit awkwardly placed, and there does not seem to be a good link between them.

Modified.

- Lines 185-186. “Nakhlites/chassignites represent melting of metasomatized mantle from early magmatic events. These meteorites could have been ejected from deep craters, or are from regions distinct from younger shergottites.” I think there are a few issues with this statement about deep craters. The logic in part of this sentence seems to be: nakhlites+chassignites are earlier than (most of) the shergottites, therefore the nakhlites+chassignites would be buried beneath the shergottites, and therefore must be from deeper in the craters. This is unnecessarily complicated, and actually goes against the evidence from other studies of martian meteorites. Firstly, we already know that the nakhlites+chassignites are from different craters to the shergottites, as these different meteorite groups have different cosmogenic exposure ages (Tomkinson et al., 2015). Secondly, we know from NASA mapping that ~1.4 Ga volcanic rocks are widely exposed at the surface of Mars (Herzog and Caffee, 2014), so there is no need to hypothesize that the nakhlites must be ejected from deep within craters.

Modified.

- Various figures: The chassignites and nakhlites are not identical, so it would be beneficial if these rock types had different symbols.

We disagree – the whole point of this manuscript is to demonstrate that they are linked by source melting. We have kept the symbols as originally intended; this is also important for the sake of clarity.

- Figure 3: not all of the lines are described in the figure caption.

Fixed.

- Figure S1: Samples that fall in the “foidite” can be classified further (Le Maitre, 2002).

They can be further described, but we don't think this is useful since the chassignites, in particular, are cumulates. For this reason we have removed the term ‘foidite’ entirely.

- Table S1: What is the “TAB” data in the last three columns? Make sure that all abbreviations or column headings are adequately described.

Thank you. Done.

In summary, the manuscript contains an important suite of analyses that will be very useful for examining martian meteorites, martian volcanism, and the geochemical evolution of that planet. However, more work needs to be done in the reporting of the methods, evaluation of the analytical results, and examination of the scientific ideas. At the moment the manuscript seems very short for Nature Communications. Therefore there is ample space for expansion of the text and number of figures, which will allow for considerable enhancement to the scope of the manuscript.

References:

Clague, D.A., 1987. Hawaiian alkaline volcanism. In: Fitton, J.G., Upton, B.G. (Eds.), *Alkaline Igneous Rocks*. Blackwell Scientific Publications, Oxford, pp. 227-252.

Demidjuk, Z., Turner, S., Sandiford, M., George, R., Foden, J., Etheridge, M., 2007. U-series isotope and geodynamic constraints on mantle melting processes beneath the Newer Volcanic Province in South Australia. *Earth and Planetary Science Letters* 261, 517-533.

Herd, C.D.K., Tornabene, L.L., Bowling, T.J., Walton, E.L., Sharp, T.G., Melosh, H.J., Hamilton, J.S., Viviano, C.E., Ehlmann, B.L., 2018. Linking martian meteorites to their source craters: new insights. LPSC, p. Abstract #2266.

Herzog, G.F., Caffee, M.W., 2014. Cosmic-Ray Exposure Ages of Meteorites. In: Davis, A.M. (Ed.), *Treatise on Geochemistry* 2nd Edition. Elsevier, Amsterdam, pp. 419-454.

Hoernle, K., Hauff, F., Werner, R., van den Bogaard, P., Gibbons, A.D., Conrad, S., Müller, R.D., 2011. Origin of Indian Ocean Seamount Province by shallow recycling of continental lithosphere. *Nature Geoscience* 4, 883-887. doi:10.1038/NNGEO1331

Le Maitre, R.W., 2002. *Igneous Rocks: A Classification and Glossary of Terms*. Second ed. Cambridge University Press, Cambridge.

Mahaffy, P.R., Webster, C.R., Atreya, S.K., Franz, H., Wong, M., Conrad, P.G., Harpold, D., Jones, J.J., Leshin, L.A., Manning, H., Owen, T., Pepin, R.O., Squires, S., Trainer, M., MSL Science Team, 2013. Abundance and isotopic composition of gases in the Martian atmosphere from the Curiosity Rover. *Science* 341, 263-266. doi:10.1126/science.1237966

Pilet, S., Abe, N., Rochat, L., Kaczmarek, M.A., Hirano, N., Machida, S., Buchs, D.M., Baumgartner, P.O., Müntener, O., 2016. Pre-subduction metasomatic enrichment of the oceanic lithosphere induced by plate flexure. *Nature Geoscience* 9, 898-903. doi:10.1038/ngeo2825

Rollinson, H., 1993. *Using Geochemical Data: evaluation, presentation, interpretation*. Prentice Hall, London.

Tomkinson, T., Lee, M.R., Mark, D.F., Dobson, K.J., Franchi, I.A., 2015. The Northwest Africa (NWA) 5790 meteorite: A mesostasis-rich nakhlite with little or no Martian aqueous alteration. *Meteoritics & Planetary Science* 50, 287-304. doi:10.1111/maps.12424

Treiman, A.H., 2005. The nakhlite meteorites: augite-rich igneous rocks from Mars. *Chemie der Erde - Geochemistry* 65, 203-270. doi:10.1016/j.chemer.2005.01.004

Udry, A., Day, J.M.D., 2018. Formation and emplacement of the cogenetic nakhlite and chassignite meteorites. LPSC, p. Abstract #1052.

We thank the reviewer for their thoughtful and constructive comments.

Comments of Reviewer 2, Professor Jeff Taylor

This paper presents a coherent and extensive set of data on Martian meteorites and uses it to present an interesting, unified view of the petrogenesis of shergottites and nakhlites/chassignites. The Martian crust is decorated by prominent, large, basaltic volcanoes, whose origin is almost always ascribed to long-lived plumes from the mantle. The paper makes a good case for shergottite magmas being analogous to the shield-build magma that produce Hawaiian volcanoes and that the nakhlites are analogous to the rejuvenated stage. The important contribution of the paper (besides the excellent data presented in the supplement) and the reason it should be published is that it draws

a coherent picture of Martian magma generation in the context of mantle plumes. Thus, I recommend publication. However, I have a few suggestions that could make the paper a bit more well-rounded, though at the risk of placing less emphasis on the integrated plume idea.

We thank Professor Taylor for his perceptive comments.

Meteorites may not be representative of the Martian crust: Shergottites and nakhlites are important sources of information about Mars, but they may not be abundant rocks. Mars Odyssey Gamma-Ray Spectrometry (GRS) shows that the crust is richer in incompatible elements (specifically K and Th) than the Martian meteorites (see Fig. 4 in Taylor et al., 2006), though they have about the same K/Th. Other parameters are similar, such as FeO (see Fig. 7 in Taylor et al., 2006). In addition, the igneous rocks at Gusev crater are richer in alkalis than are the shergottites (see McSween et al., 2006), although we do not have the complete set of elements that appears in Day et al. Gusev does house some olivine basalts, too. Perhaps it would be more complete to point out the differences in bulk crustal compositions from GRS and the nature of Gusev (and Gale, too) rocks compared to Martian meteorites. What could be the relation of the bulk crust and the Gusev rocks to the plume idea?

An important point and one that we consider in more detail in the manuscript.

NWA 7034 etc. seems very different from the shergottites. The Black Beauty breccia opens up other questions. It has alkalic, gabbroic clasts in it. Do these relate to one of the plume components, or do they represent another form of magma production in Mars? U-Pb and Sm-Nd dating indicates that they are very old, 4.42 Ga, so one might argue that they formed by a mechanism different from plumes, if the plume phase of Martian mantle evolution started later. I have no answers, but it could be useful to mention these rocks. Some of the key elements to test the fit to the plume idea have been analyzed and presented in the Humayan et al. paper on NWA 7034.

Thank you. We consider this in more detail in the manuscript.

No volcano in Gusev. To return to the Gusev rocks again, if the olivine basalts and alkalic basalts indicate a main shield stage and a rejuvenation stage indicate a plume-related process, where is the volcano?

We are not sure how this can be reconciled with the alkalic nature of Gusev Crater rocks, but we consider this nonetheless.

Ferrobasalts on Earth. Justin Filiberto (2008) wrote an interesting paper discussing terrestrial analogs for the shergottites. He drew attention to ferrobasaltic rocks in the oldest eruptives in flood basalt provinces and as effusive flows and intrusive sills in some ocean floor locales such as Kola Peninsula. An important point is that these flows are associated with voluminous tholeiitic flows, hence perhaps making the connection between rejuvenation and shield-building stages. In addition, ideas for their petrogenesis include plumes and creation of a metasomatized mantle region (summary in section 3.1 of Justin's paper). These rocks might enhance the argument for a plume-style model. Or might show it is not always needed to explain a suite of rocks.

We have modified the manuscript accordingly in response to this comment.

Heterogeneous mantle produced during magma ocean crystallization, and mantle sources in general. A conventional idea is that the two shergottite sources, enriched and depleted, formed early in the history of Mars, by processes in the magma ocean. Does the plume model rule this out? Does it apply to initial creation of mantle sources? One can imagine a heterogeneous mantle still having plumes that behave something like described in the Day et al. manuscript. Also, others have drawn attention to diversity of mantle sources, including some compositional features that require missing end members. For example, see Fig. 6c in Taylor et al. (2008, in Jim Bell's Big Book), shows that all Martian meteorites are on or above the terrestrial (= chondritic) line on a plot of Ba/La vs La. Does this have anything to do with the plume idea? Does it weaken the hypothesis? (I doubt it, but it is worth pointing out this complexity. (Black Beauty falls on the terrestrial line.).

No, it doesn't weaken the idea, but we include the discussion.

Filiberto, J. (2008) Similarities between the shergottites and terrestrial ferropicrites. *Icarus*, 197, 52-59.

Humayun, M., Nemchin, A., Zanda, B., Hewins, R. H., Grange, M., Kennedy, A., Lorand, J.-P., Göpel, C., Fieni, C., Pont, S., and Deldicque, D. (2013) Origin and age of the earliest Martian crust from meteorite NWA 7533. *Nature*, v. 503, p. 513-516. doi:10.1038/nature12764.

McSween et al. (2006) Alkaline volcanic rocks from the Columbia Hills, Gusev crater, Mars. *Geophys. Res.* 111, doi:1029/2006JE002698.

Taylor, G. J., et al (2007), Bulk Composition and Early Differentiation of Mars, *J. Geophys. Res.* 111, E03S10, doi:10.1029/2005JE002645.

Taylor, G. J., McLennan, S.M., McSween, H. Y., Jr., Wyatt, M. B., and Lentz, R. C. F. (2008) Implications of observed primary lithologies. In *The Martian Surface: Composition, Mineralogy, and Physical Properties* (J. F. Bell, ed.), 501-513. Cambridge University Press.

Comments of Reviewer 3, Anonymous

It is challenging to reconstruct the major petrogenetic processes occurring on a planet with only a limited and probably biased set of samples. This is definitely the case with Mars, and authors of the submitted manuscript rightly claim that no satisfactory unified model has been put forth to explain the petrogenesis of shergottites, nakhlites, and chassignites, the SNC suite of meteorites. Using a new, internally consistent dataset of trace element concentrations from 25 shergottites, together with existing nakhlite data and a simple melt model, the authors draw on analogies between shergottites and the Hawaiian main shield basalts and between nakhlites (and chassignites) and the products of Hawaiian "rejuvenated" volcanism to explain the petrogenetic relationship among SNC members. This is a novel and very interesting concept and it makes predictions about the spatial relationships between these rock types that could be tested by future missions to volcanic edifices on Mars. Particularly thought-provoking is the focus on plume magmatism, to draw petrogenetic parallels between a tectonically active Earth on the one hand and Mars, with its lack of plate tectonics on the other. The authors focus mainly on only three of the suite of 41 analyzed trace elements, ie., Nb, Zr, and Y. These have the

advantage of being sensitive to certain igneous differentiation processes, while being fluid immobile and thus expected to be intact in meteorites known to have undergone hot- or cold desert weathering and alteration. On the basis of higher Nb/Y at a given Zr/Y of nakhlites and rejuvenated Hawaiian lavas, and the fact that the later have a contribution from a metasomatized depleted source, the authors suggest (reasonably) that a similarly metasomatized source can explain the nakhlite (and chassignite) compositions. Despite the novel ideas presented here, I have some misgivings outlined in my main comments below that prevent me from recommending this for immediate publication. I think these would be easily addressed in a larger, more detailed article submitted elsewhere.

We thank the reviewer for their constructive comments. We feel that the issue of 'a larger, more detailed article' was easily rectified by the longer word limit for Nature Communications.

* Ruling out effects of surface processes on Earth and Mars: Many martian meteorites have been affected by terrestrial weathering, alteration, and contamination (the latter meaning addition of fine-grained material during terrestrial residence or sawing). These will probably have little effect on Nb, Zr, and Y, but it would be good to demonstrate this. Sawing off fusion crust is a good start, but how can the other effects be monitored and determined to be insignificant? When adding isotope ratios and other trace element constraints to the model (see below), significant effects of surface processes need to be confidently ruled out first.

This is a very similar comment to Reviewer 1. We now add a much more comprehensive discussion to show that this is not an important effect.

* Analytical methods: The digestions were done at low pressure on a hotplate. Therefore some refractory HFSE-bearing phases might not have been completely digested. It needs to be shown (perhaps using other digestion methods that attack HFSE minerals) that this has not affected the axes of the main Fig 3 plot. To their credit, the authors have supplied a sufficient number of geologic standard analyses to allow proper assessment of data quality.

We do this with confidence – see the analytical methods.

* The partial melting model needs to be much better documented and explained than it is. This means including 1) a table of the actual partition coefficients used (not just a catch-all reference to a large database plus 3 specific papers, 2) the calculations used to define model gridlines and melting arrows on Fig. 3, and 3) a realistic assessment of error propagations. It is difficult to see the primitive and depleted mantle models under the data points on Fig. 3. The difference between dashed lines and solid lines also needs to be explained.

Agreed and modified.

* The presented novel hypothesis of the Hawaiian rejuvenated volcanism analog departs significantly from earlier petrogenetic models (and this a good thing). A successful hypothesis explains all existing data, not just a subset of trace elements. Although the authors qualitatively discuss major element and isotopic constraints in the context of their model, their idea would be more convincing if backed up by _quantitative_ constraints

from existing isotope data, as well as other trace- and major element data. The authors might also want to consider oxygen fugacity constraints (e.g., Herd, C.D.K., 2003. The oxygen fugacity of olivine-phyric martian basalts and the components within the mantle and crust of Mars. *Meteoritics & Planetary Science* 38, 1793–1805.) The authors should also think about how far removed individual meteorites are from parental melt compositions (via fractional crystallization, crystal accumulation, or assimilation). This issue was highlighted early on: Longhi, J., 1991. Complex magmatic processes on Mars—Inferences from the SNC meteorites. *Proc. Lunar Planet Sci.* 21, 695–709.

Modified and included to the text.

Specific comments:

24: what does “lower coupled Nb/Y-Zr/Y” mean? Lower Nb/T at a given Zr/Y.

Fixed.

68-69: “depleted Rb/Sr and Sm/Nd isotopic compositions” – I know what is meant, but Rb/Sr and Sm/Nd are primarily element ratios, not isotopic compositions. I recommend “depleted Sr and Nd isotopic compositions”

Okay, changed.

72: Again, be careful with the use of “enriched” and “depleted” with element ratios. Here “Sm/Nd enriched” does not make sense. “Enriched” means incompatible element enriched, i.e., a lower Sm/Nd compared to the reference reservoir. To explain ^{142}Nd enrichments however, the Sm/Nd had to be higher than the reference reservoir.

Modified.

Adjective-noun & subject-verb agreement:

21: “dataset...reveals”

157: “basalt compositions”

165: “differences...suggest”

178: “Nakhlites come”

382: “a source...is required”

406: “plot the diagram”

Changed

Style suggestions: Avoid “and/or.” Logically, “or” includes the option of “and” and is thus not equivalent to “either...or” (Line 52)

Modified.

Avoid ambiguous or non-committal slash “/” constructions. Does the slash denote a ratio, alternative choices, “or,” or “and”? Line 140 gives a particularly ambiguous example: “shield stage or pre-/post-shield stage” covers all possibilities.

Modified.

Replace “nakhlites/chassignites” with “nakhlites and chassignites” at first and perhaps with “NC” afterwards for brevity.

Modified.

Avoid anthropomorphisms. Rocks do not “experience” and have not “seen” anything. (Lines 42, 179)

Fixed.

FIGURE 2: Cite data sources clearly; Note that none of the Nakhilite data are from the present study.

Changed.

FIGURE 3: With the exception of the metasomatised source model, the models in the figure are difficult to see or follow. What is the significance of the grey dotted lines?

Changed.

339: “martial”->“martian”

Indeed – changed.

SUPPLEMENTARY MATERIALS:

55, 356: Both assemblages should be clarified by adding the word “and” before “spinel or garnet”

Modified.

FIGURE S1: The caption should more directly label the data sources. According to the caption of Table S1, no major element data are from the current study.

Fixed.

TABLE S1: According to the table caption, the major element data for the shergottites, and the major- and trace element data for nakhlites are being primarily reported in two papers submitted to EPSL. It seems inappropriate to include the data in the current data table until those papers have been accepted. Most major element concentrations are displayed at a reasonable number of significant digits (which happens to correspond to two decimal places for the major elements). However, no trace element data measured by concentration reference ICP-MS should be displayed to more than 3 significant digits. It is especially not warranted when the external reproducibility is percent-level or worse (“better than 6%” for most elements; >10% for Mo, Te, and Se.) It would also be good to round off the values in the table to the correct number of significant digits rather than to maintain 15 digits of precision in each cell.

We address all these comments in the manuscript.

For facilitating future references to this data set, it would be good to provide each analysis with a unique ID number or at least label replicates a, b, c, etc.

Okay – modified.

What do the “TAB” columns at the end mean? Some SiO₂ contents (NWA 7721, 6342) were measured by difference assuming 100% total. Why?

These are all explained in the Supplement.

Reviewers' Comments:

Reviewer #1:

Remarks to the Author:

Day et al have made many modifications to this manuscript, and I think it is now considerably improved and more balanced. There are still some outstanding issues, but should all be easy to address. My comments are listed below:

Line 40: "pre-shield, shield and post-shield erosional and rejuvenated stages". The first 'and' is not required, and should be replaced by a comma.

Lines 80-81: I think the sentence would benefit by adding 'respectively' to the end of the sentence, i.e., "analogous to Hawaiian main shield and rejuvenated volcanism, respectively."

Lines 86-87: This sentence states the number of shergottites analysed. As Table S1 now also includes the data for the chassignite and nakhlite meteorites, I think it would also be beneficial to also report the numbers of nakhlite and chassignite meteorites that were analysed.

Lines 89-91: It would be useful to refer to Figure 1 at the end of this sentence: "Shergottites range from micro-basalt to basaltic compositions, with enriched shergottites having elevated total alkalis (Na₂O + K₂O) relative to intermediate or depleted shergottites (Figure 1)."

Lines 105-107: "Sample NWA 6963 is exceptional in that it has the major element geochemical composition of a shergottite, but the incompatible trace element composition of a nakhlite." I commented on this sentence in the first version of the manuscript (it was formerly on lines 92-93), but as far as I can see, the text remains unchanged, although the rebuttal document states this sentence has been modified. The statement needs to be supported with evidence, i.e.:

- 1) Please cite Figure 4, as this figure clearly shows that trace element data from NWA 6963 overlaps with the nakhlites.
- 2) Show evidence that NWA 6963 has "the major element geochemical composition of a shergottite". The data on Figure 2 are inconclusive, as the shergottite and nakhlite analyses overlap. Is there another figure or paper that the authors could cite to confirm this statement?
- 3) Alternatively, has NWA 6963 been analysed for Sr and Nd isotopes? In which case the sentence could be modified to "Sample NWA 6963 is exceptional in that it has the isotopic composition of a shergottite, but the incompatible trace element composition of a nakhlite." (I did a quick literature search, and couldn't find Sr or Nd isotopic data for NWA 6963, but the authors may be aware of data that I am not familiar with). Sr and Nd isotopic data demonstrating overlap with the shergottites would much more conclusive than evidence from major element results – as the latter are considerably affected by the mineralogy – especially as NWA 6963 is a cumulate (Filiberto et al, accepted 2018, JGR).

Line 119 (and elsewhere): check that the "±" symbol always has a space on either side. (Most instances have been corrected, but there are still a few formatting errors remaining.)

Line 243: the statement "weathering or fusion crust surfaces were removed" could benefit from a bit more detail, e.g., "were removed by wire saw" or "were removed by wire saw followed by manual picking under a binocular microscope" or similar.

Line 243: In the original review I commented that the authors should "state the total mass of rock that was crushed in order to produce the powders used for major and trace element analysis". This has not yet been added, and is an important aspect of the methodology that needs to be included so that this study can be replicated. The mass of rock that was ground to powder is also crucial to address the questions about potential sample heterogeneity affecting the chemical results. Lines 130-150 are good, and go some way to address the comments that reviewer 3 and I made on the initial submission, but the methods still require information on the mass powdered.

Lines 248, 251: reporting that the acids used were “concentrated” is insufficient detail to ensure the methodology can be replicated. The authors should report these concentrations in Molality, Molarity, or similar quantitative measure.

Line 250: To help address one of the comments by Reviewer 3 (about the potential for non-total dissolution), it may be useful to add something like “The procedures used were identical to those successfully and routinely used to fully dissolve terrestrial mafic igneous rocks.”

Lines 265-266: quantify (in %) what is meant by the non-exact terms ‘excellent’ and ‘good’.

Lines 270-272: “A caveat is that limited sample masses can lead to a mode-effect, where non-representative volumes of rock are chosen, resulting in greater variability in chemical measurements” To the end of this sentence, possibly add “as described in the discussion” in order to direct the reader to the more comprehensive evaluation of this topic that is located in the discussion.

Line 304: change to “(for depleted shergottites)”

Line 305: change to “(for intermediate and enriched shergottites)”

Line 312: why is the chemical formula for rutile provided, while the chemical formulae for other minerals are not? I suggest removing the chemical formula for rutile.

Lines 334-338 and Table S2: The ages of the nakhlites. In the revised submission, Day et al. have included Table S2 on the ages of the nakhlites. Table S2 is useful, but there are several points that require improvement:

- 1) Each age should be matched to its source reference, which is crucial for data traceability (good examples are Tables 2 and 3 in Nyquist et al. 2001. Space Science Reviews 96, 105-164).
- 2) The crystallisation age and the exposure age are usually measured by different techniques – e.g., the crystallization age by Ar/Ar, while the exposure age is often determined by ^{38}Ar or other noble gases, not Ar/Ar. Extra information needs to be added to the table, for example, an additional column with the method used for the exposure age.
- 3) The statement that “Uncertainties are typically reported as 1 St. Dev.” is not accurate, as many of the ages cited had analytical uncertainties reported as 2 Std. Dev. There needs to be additional information added, e.g., specifying which uncertainties are 1 Std. Dev. vs. 2 Std. Dev., or recalculating the uncertainties to be consistently 1 or 2 Std. Dev.
- 4) Could the list of meteorites be arranged in some order (e.g., alphabetical)? At the moment it seems quite random.

From the data in Table S2 the authors calculate a “grand average” age of 1340 ± 40 Ma. If they use this type of calculation, in the manuscript or supplementary table they need to justify why they have calculated a single average age with a large uncertainty. Such a calculation is problematic for the chassignites and nakhlites, because several meteorites have statistically distinct ages (e.g., Gov Valad 1330 ± 10 Ma; Lafayette 1330 ± 15 Ma; Nakhla 1383 ± 7 Ma; Y000749 1415 ± 8 Ma; Table S2), which would indicate a range of formation ages for the various meteorites.

In the rebuttal, Day et al. comment “the issue raised of multiple flows from Ar-Ar age data for nakhlites alone is not so straight-forward. Independent evidence for multiple lava flows from petrological constraints (e.g., Udry & Day, in review) are not always correlated with new age data for nakhlites. The evidence for a range of ages related to multiple eruptions is not as straightforward as presented in Cohen et al. (2017), for example.”

What is meant by “petrological constraints (e.g., Udry & Day, in review) are not always correlated

with new age data for nakhlites"? Obviously the ages and petrological constraints need to be geologically compatible, but is there a reason why the ages should be correlated with the petrological constraints? A volcano can change geochemically and petrologically in complex ways through time (e.g., Rhodes & Vollinger G3, (2004) doi:10.1029/2002GC000434). Why is the "evidence for a range of ages related to multiple eruptions is not as straightforward as presented in Cohen et al. (2017)"? There needs to be evidence for this statement.

Evidence for multiple igneous units is not just from Ar/Ar ages, there are also petrographic, geochemical, and isotopic arguments for multiple igneous units, for example Jambon et al. (2016, GCA doi:10.1016/j.gca.2016.06.032) and Righter et al. (2016) LPSC Abstract # 2780. In addition, Udry and Day (2018, LPSC Abstract #1052) state: "According to textural analyses, we suggest that nakhlites were emplaced in different lava flows or sills." To me, evidence of different lava flows or sills is entirely consistent with different formation ages, therefore there is no conflict between chronology and chemistry, and that the range of ages determined by geochronology should be reported.

This paper by Day et al. obviously isn't focussed on the age of the nakhlites and chassignites, so I don't think or want the manuscript or supplementary material to have to get bogged down in discussion of that topic. Ideally, the most precise and accurate age determined from chronology should be used for each meteorite (i.e., as is standard practice for chronology of terrestrial rocks), but if Day et al. provide evidence why that is not applicable, then would a statement like this suffice at Line 337? "The nakhlites and chassignites have yielded ages ranging from ca. 1300 to 1400 Ma (Table S1). For the purposes of calculating the initial Sr and Nd isotopic compositions of these meteorites, we used an average of all of the ages for these meteorites, of 1340 ± 40 Ma."

Elsewhere in the manuscript, the age of the nakhlites and chassignites should only appear with some indication of uncertainty (e.g., they should not be cited as an exact "1340 Ma", which appears on lines 16, 61, 70, and 222). A possibility could be: "nakhlites and chassignites (ca. 1300-1400 Ma)".

Various figures (especially 2-5):

Initial review: The chassignites and nakhlites are not identical, so it would be beneficial if these rock types had different symbols.

Rebuttal: We disagree – the whole point of this manuscript is to demonstrate that they are linked by source melting. We have kept the symbols as originally intended; this is also important for the sake of clarity.

Response to rebuttal: Day et al are indeed correct that the chassignites and nakhlites are very similar chemically, and I agree with their interpretation in this manuscript that they are linked by source melting. However, I will re-iterate my earlier point that these rocktypes are not identical – that's why they have been classified into different groups. In particular, the nakhlites have abundant clinopyroxene, while the chassignites are rich in olivine. These mineralogical differences will affect the geochemistry (e.g., especially MgO on Figure 3). I think that the point that the nakhlites and chassignites are linked by source melting would actually be enhanced by having (subtly) different symbols for the two groups. A very easy option would be to keep using stars with a black outlines as the symbol for both the chassignites and the nakhlites. To distinguish the chassignites from the nakhlites, one of these groups (possibly the chassignites, as there are fewer of them) could have the inside of the star shaded a different colour. This option would not compromise the clarity of the figures, but would enhance the science.

Figure 2: the blue dots should also be explained, either in the legend or the caption.

Figure 4: the caption needs the statement "Analytical uncertainties are smaller than symbols".

Figure 5: the grey dashed lines still need to be explained.

Table S1: ages are listed for the shergottites, but no sources for these ages are reported.

Table S1: it isn't immediately apparent why are there three columns for the Total Analytical Blanks. There needs to be some more explanation of this aspect of the data.

Table S1 and S2: the supplementary files provided for this review were in the format of pdf portrait pages. This resulted in two major issues: 1) all pages except the first page lacked the column on the left that describes what was on each row, and 2) any text on the header and footer lines were split over multiple pages. This may be a formatting issue that has occurred during the journal electronic submission process, but these issues are obviously not ideal for readability, and need to be resolved before publishing.

Reviewer #2:

Remarks to the Author:

This is the second round of reviews, but for Nat Comm, which allows for longer papers. I think the authors have answered and in some cases rebutted sensibly the reviewer comments on the first version. The paper is interesting and presents an important unifying concept, put into a global perspective. It puts the compositions of Martian meteorites into the framework of a plume-driven planet, which is consistent with geological observations and geophysical modeling. I predict that it will be widely cited and will be the springboard for numerous studies. Every new Martian igneous rock we analyze will be put into the same framework to see if it fits.

Reviewer #3:

Remarks to the Author:

Reviewer #3 from the last round; Please refer to my previous summary, which won't be repeated here. .

This manuscript, which provides new chemical data and a novel petrogenetic view of the Martian meteorite (SNC) suite is much improved over the first round. Below I list my main comments from last time and how the authors have responded.

1. TERRESTRIAL SURFACE PROCESSES AFFECTING METEORITES (sufficiently addressed):

Spikes in fluid-mobile elements (Ba, U, Sr) in a few of depleted shergottite patterns are suggestive of terrestrial weathering or alteration. However, the authors now better explain why the HFSE should be relatively invulnerable to fluid-mediated alteration and weathering.

Addition of solid material was not considered: Contamination by "fluid-immobile" elements is still possible however if water infiltrating among cracks brings terrestrial dust with it, i.e., not dissolved, but in fine suspension. It would be reassuring to see that possibility excluded by using the trace element data set, possibly combined with isotopic data. One could, e.g., take the most depleted sample in each group and model the addition of a loess-like component to such that the maximum HFSE concentrations are reached. If either the resulting trace element pattern or isotope composition does not fit those of the actual high-HFSE samples, then possible contamination by terrestrial dust can be eliminated.

In short, it should be easy to test —with existing data— for the presence of a terrestrial dust contaminant. I think the authors can decide if they want to do this or not.

2. ANALYTICAL METHODS: SAMPLE DIGESTION (not sufficiently addressed)

In my previous review I had mentioned that hot-plate digestions often do not digest refractory HFSE-bearing minerals. If there is a systematic difference in the types or amounts of refractory

minerals present in nakhlites and chassignites vs. shergottites, then hot-plate digestions might cause a bias.

I suspect that this is not the case with the current dataset, but it would be easy enough to test by doing high-pressure digestions (autoclave bombs) of just a few samples and comparing their positions to hot-plate digestions in Fig. 5.

In the rebuttal, the authors state they do digestions "with confidence" adding in line 250 of the revised manuscript: "Acid attack led to complete dissolution of rock samples to generate clear solutions, with no remaining solid material." - I am sceptical that Ca+Mg-bearing samples went into complete solution in a 4:1 HF-HNO₃ mixture at the given sample-to-acid ratios. Also, simply taking up the samples in concentrated HNO₃ will generally not destroy fluorides, but several dry-downs with HNO₃ might work. Is that what was done?

In summary think the authors should provide solid evidence that the hotplate digestions were effective, especially for refractory Zr-rich minerals, which would affect their Nb/Y vs. Zr/Y diagram.

3. PARTIAL METLING MODEL (not sufficiently addressed):

The authors have greatly improved their explanation of the modelling in the methods. However, Figure 5a is still a bit cluttered (black model lines obscured by data), and the authors still do not list specific partition coefficients used in their model. Line 287: Citing a partition coefficient database is not helpful unless specific data are cited. More useful to those trying to reproduce the models here would be a simple table of the actual partition coefficients (or ranges thereof) that were actually used and their specific publication reference. The reason is that among different studies, absolute partition coefficients tend to vary much more than RATIOS of partition coefficients; it would be good to be able to assess (through the table) whether the authors have used absolute partition coefficients that are internally consistent with ratios thereof.

4. REJUVENATED VOLCANISM ON MARS (sufficiently addressed):

Last time I had wondered whether this idea, supported by Nb/Y vs. Zr/Y and qualitative isotopic arguments, could benefit from quantitative modelling using existing isotope data. I had meant whether the implied sources of shergottites (S) and nakhlites (N) plus chassignites (C) could be linked by not only trace element patterns, but also by the isotopic evolution of their sources. OR perhaps one could predict what the isotope compositions of shergottites extracted from the N+C source would be. The former is probably difficult however given the age gap between S and N+C, and both are probably hampered by unknowns regarding the metasomatic agent. For this reason, I think it is fine for the authors keep isotopic classifications qualitative (e.g., long-term depleted vs. enriched sources) as they currently stand.

Line-by-line specific comments; (*) indicates comments repeated from first review.

18: "lithology" means "the study of rocks," and thus should not be used to indicate e.g., "rock type" or "lithologic unit," even though such usage is rampant in the literature. I recommend changing "distinct lithologies" to "distinct rock types" or "distinct rock lineages"

21(*): "lower coupled Nb/Y-Zr/Y" - as I alluded to in my previous review, this wording may be compact, but is vague. What is meant here is "lower Nb/Y at a given Zr/Y." As written, a reader might think that shergottites are simply further down toward the Y-intercept on the same trend and N+C.

21: "Shergottites have lower coupled Nb/Y-Zr/Y than nakhlites or chassignites, a trend that is nearly identical to Hawaiian main shield and rejuvenated volcanism on Earth." I recommend changing "trend" to "pattern" (because there are two trends, not one) AND adding ", respectively,"

after volcanism.

35(*) "Mars..experienced" - avoid anthropomorphisms. "undergone" or "had"

41: "due to" (adjectival) -> "owing to" (adverbial; modifying "have not been recognised")

59: Figure 1: Just an observation: These samples are wound back to different times in the past, reflecting instantaneous isotopic snapshots of sources that evolved through time. As such, the figure blurs the time dimension so relations of among source reservoirs at any given point in time are difficult to make from this diagram. It does serve its purpose however in sorting sources into broadly depleted or enriched sectors.

64: "nakhlite and chassignites" (add an "s" to the former)

68 Delete hyphen between late and incompatible.

82: "Mars experiences" - anthropomorphism. Rerword.

86: "compare these data with THOSE obtained for nakhrites and chassignites.."

114: "shergottites and nakhrites and chassignites" - I think in response to my earlier comment, a slash was replaced by an "and" but the grouping is not clear. Perhaps replace the first "and" with "vs."

117: add "s" to "nakhrite"

119: again eliminating a slash made the grouping unclear. Suggest changing the second "and" to "plus" or similar.

139-150: Inter-element ratios are "unaffected" by the mode effect. Define "unaffected" — no difference outside of analytical uncertainty? Within X%? Also: Would it make sense to use the mean values of replicate analyses to gain a more representative value?

140: "meteoritics" - Do not make nouns out of adjectives by adding "s" (meteorites)

141: "due to" -> "owing to" (adjectival vs adverbial modifier; the verb here being "is")

181(*) "pre-/post-shield stage" - avoid such vague slash constructions.

187: "based on" (adjectival modifier) —> "on the basis of" (adverbial modifier), the modified verb being "are observed"

188: "similar TO — although significantly larger than— flexural bulges" (add "to" and offsetting m-dashes or commas)

201: "Models of 5-10% partial melting of primitive mantle reproduce Hawaiian shield compositions, and of depleted mantle reproduce mid-ocean ridge basalt compositions" - awkward sentence structure.. perhaps replace "and of depleted" with "whereas those of depleted"

214: is "tectonic" the right word here? This is more of a deeper geodynamic difference, right? Stirring and tectonic features are the results of convection. Convection and stirring occur in both plate tectonic and stagnant lid scenarios. Some studies, however, e.g., Debaille et al. (2013, EPSL 273), show that stagnant lid scenarios may better preserve isotopic variability than plate tectonic ones.

222 "shergottite crystallization ages (165-2400 Ma) generally preclude a direct link between nakhrites and chassignites." In the previous version of the manuscript, a direct link between younger shergottites and nakhrites was precluded by their age differences, which made sense to

me. Now the precluded link is between nakhlites and chassignites — on the basis of shergottite ages, which does not make sense to me. Perhaps the authors meant: “..generally preclude a direct link to nakhlites and chassignites.”

224(*): “from a source that has seen prior melt depletion” - sources do not “see” anything. Avoid anthropomorphism.

226: Observations of 2370 to 2400 Ma shergottite magmatism on Mars supports continued and persistent plume magmatism for at least two billion years” — Well the ages quoted here just mark the older end of the magmatism interval. Perhaps quote and cite the full shergottite age range here.

231: “The distinctive nature of martian meteorites [as compared to] remotely sensed martian surface samples” OR “The distinctive natures of martian meteorites [and] remotely sensed martian surface samples”

235: “offering a mechanism for explaining” = “potentially explaining”

279: Consider rewording: “correction for non-systematic laboratory bias” - A bias is a systematic error. Biases between labs can be corrected using data from international standards published alongside the unknowns. What cannot be corrected are “non-systematic” (random) errors within a lab’s data. Replicate data of standards from a single laboratory give the external reproducibility and thus is a measure of the magnitude of the random error in a lab’s data.

284: “the source[s] of rejuvenated lavas and nakhlite[s] and chassignites versus [the sources] of Hawaiian main shield stage lavas and shergottites.

337: “based on” -> “on the basis of”

Table S1:

There are still concentrations and blanks listed in this table that have 4 and 5 significant digits, which is unrealistically precise given the external reproducibility.

Table S2:

Move the method column next to the crystallisation ages. The given methods are not for exposure ages.

“standard deviation” is usually abbreviated as S.D., SD, or s.d.

Comments of Reviewer 1, Anonymous

Day et al have made many modifications to this manuscript, and I think it is now considerably improved and more balanced. There are still some outstanding issues, but should all be easy to address. My comments are listed below:

Line 40: “pre-shield, shield and post-shield erosional and rejuvenated stages”. The first ‘and’ is not required, and should be replaced by a comma.

Done.

Lines 80-81: I think the sentence would benefit by adding ‘respectively’ to the end of the sentence, i.e., “analogous to Hawaiian main shield and rejuvenated volcanism, respectively.”

Added.

Lines 86-87: This sentence states the number of shergottites analysed. As Table S1 now also includes the data for the chassignite and nakhlite meteorites, I think it would also be beneficial to also report the numbers of nakhlite and chassignite meteorites that were analysed.

Added.

Lines 89-91: It would be useful to refer to Figure 1 at the end of this sentence: “Shergottites range from micro-basalt to basaltic compositions, with enriched shergottites having elevated total alkalis ($\text{Na}_2\text{O} + \text{K}_2\text{O}$) relative to intermediate or depleted shergottites (Figure 1).”

Figure 2 – added.

Lines 105-107: “Sample NWA 6963 is exceptional in that it has the major element geochemical composition of a shergottite, but the incompatible trace element composition of a nakhlite.” I commented on this sentence in the first version of the manuscript (it was formerly on lines 92-93), but as far as I can see, the text remains unchanged, although the rebuttal document states this sentence has been modified. The statement needs to be supported with evidence, i.e.:

1) Please cite Figure 4, as this figure clearly shows that trace element data from NWA 6963 overlaps with the nakhlites.

2) Show evidence that NWA 6963 has “the major element geochemical composition of a shergottite”. The data on Figure 2 are inconclusive, as the shergottite and nakhlite analyses overlap. Is there another figure or paper that the authors could cite to confirm this statement?

3) Alternatively, has NWA 6963 been analysed for Sr and Nd isotopes? In which case the sentence could be modified to “Sample NWA 6963 is exceptional in that it has the isotopic composition of a shergottite, but the incompatible trace element composition of a nakhlite.” (I did a quick literature search, and couldn’t find Sr or Nd isotopic data for NWA 6963, but the authors may be aware of data that I am not familiar with). Sr and Nd isotopic data demonstrating overlap with the shergottites would be much more conclusive than evidence from major element results – as the latter are considerably affected by the mineralogy – especially as NWA 6963 is a cumulate (Filiberto et al, accepted 2018, JGR).
We have modified as best we can. Since the statement is correct, based on our data, we see no reason to modify it.

Line 119 (and elsewhere): check that the “±” symbol always has a space on either side. (Most instances have been corrected, but there are still a few formatting errors remaining.)

We have checked through the manuscript. These issues can also be corrected in proof stage.

Line 243: the statement “weathering or fusion crust surfaces were removed” could benefit from a bit more detail, e.g., “were removed by wire saw” or “were removed by wire saw followed by manual picking under a binocular microscope” or similar.

Added.

Line 243: In the original review I commented that the authors should “state the total mass of rock that was crushed in order to produce the powders used for major and trace element analysis”. This has not yet been added, and is an important aspect of the methodology that needs to be included so that this study can be replicated. The mass of rock that was ground to powder is also crucial to address the questions about potential sample heterogeneity affecting the chemical results. Lines 130-150 are good, and go some way to address the comments that reviewer 3 and I made on the initial submission, but the methods still require information on the mass powdered.

Masses were greater than 0.5g in all cases and >1-2g in most cases. We are not sure exact masses for some of these rocks could be replicated, given these samples are so difficult to get in some cases. We add more text to this effect.

Lines 248, 251: reporting that the acids used were “concentrated” is insufficient detail to ensure the methodology can be replicated. The authors should report these concentrations in Molality, Molarity, or similar quantitative measure.

Added.

Line 250: To help address one of the comments by Reviewer 3 (about the potential for non-total dissolution), it may be useful to add something like “The procedures used were identical to those successfully and routinely used to fully dissolve terrestrial mafic igneous rocks.”

Good point – thank you.

Lines 265-266: quantify (in %) what is meant by the non-exact terms ‘excellent’ and ‘good’.

Modified wording – the plots are available in Tait and Day (2018) EPSL – we should not reproduce them here.

Lines 270-272: “A caveat is that limited sample masses can lead to a mode-effect, where non-representative volumes of rock are chosen, resulting in greater variability in chemical measurements” To the end of this sentence, possibly add “as described in the discussion” in order to direct the reader to the more comprehensive evaluation of this topic that is located in the discussion.

Thank you – added.

Line 304: change to “(for depleted shergottites)”

Done.

Line 305: change to “(for intermediate and enriched shergottites)”

Done.

Line 312: why is the chemical formula for rutile provided, while the chemical formulae for other minerals are not? I suggest removing the chemical formula for rutile.

Removed.

The statement below is all one comment on the ages of nakhlites – please see our response, below:

Lines 334-338 and Table S2: The ages of the nakhlites. In the revised submission, Day et al. have included Table S2 on the ages of the nakhlites. Table S2 is useful, but there are several points that require improvement:

1) Each age should be matched to its source reference, which is crucial for data traceability (good examples are Tables 2 and 3 in Nyquist et al. 2001. Space Science Reviews 96, 105-164).

2) The crystallisation age and the exposure age are usually measured by different techniques – e.g., the crystallization age by Ar/Ar, while the exposure age is often determined by ^{38}Ar or other noble gases, not Ar/Ar. Extra information needs to be added to the table, for example, an additional column with the method used for the exposure age.

3) The statement that “Uncertainties are typically reported as 1 St. Dev.” is not accurate, as many of the ages cited had analytical uncertainties reported as 2 Std. Dev. There needs to be additional information added, e.g., specifying which uncertainties are 1 Std. Dev. vs. 2 Std. Dev., or recalculating the uncertainties to be consistently 1 or 2 Std. Dev.

4) Could the list of meteorites be arranged in some order (e.g., alphabetical)? At the moment it seems quite random.

From the data in Table S2 the authors calculate a “grand average” age of 1340 ± 40 Ma. If they use this type of calculation, in the manuscript or supplementary table they need to justify why they have calculated a single average age with a large uncertainty. Such a calculation is problematic for the chassignites and nakhlites, because several meteorites have statistically distinct ages (e.g., Gov Valad 1330 ± 10 Ma; Lafayette 1330 ± 15 Ma; Nakhla 1383 ± 7 Ma; Y000749 1415 ± 8 Ma; Table S2), which would indicate a range of formation ages for the various meteorites.

In the rebuttal, Day et al. comment “the issue raised of multiple flows from Ar-Ar age data for nakhlites alone is not so straight-forward. Independent evidence for multiple lava flows from petrological constraints (e.g., Udry & Day, in review) are not always correlated with new age data for nakhlites. The evidence for a range of ages related to multiple eruptions is not as straightforward as presented in Cohen et al. (2017), for example.”

What is meant by “petrological constraints (e.g., Udry & Day, in review) are not always correlated with new age data for nakhlites”? Obviously the ages and petrological constraints need to be geologically compatible, but is there a reason why the ages should be correlated with the petrological constraints? A volcano can change geochemically and petrologically in complex ways through time (e.g., Rhodes & Vollinger G3, (2004) doi:10.1029/2002GC000434). Why is the “evidence for a range of ages related to multiple eruptions is not as straightforward as presented in Cohen et al. (2017)”? There needs to be evidence for this statement.

Evidence for multiple igneous units is not just from Ar/Ar ages, there are also petrographic, geochemical, and isotopic arguments for multiple igneous units, for example Jambon et al. (2016, GCA doi:10.1016/j.gca.2016.06.032) and Richter et al. (2016) LPSC Abstract # 2780. In addition, Udry and Day (2018, LPSC Abstract #1052) state: “According to textural analyses, we suggest that nakhlites were emplaced in different lava flows or sills.” To me, evidence of different lava flows or sills is entirely consistent with different formation ages, therefore there is no conflict between chronology and chemistry, and that the range of ages determined by geochronology should be reported.

This paper by Day et al. obviously isn’t focussed on the age of the nakhlites and chassignites, so I don’t think or want the manuscript or supplementary material to have to get bogged down in discussion of that topic. Ideally, the most precise and accurate age determined from chronology should be used for each meteorite (i.e., as is standard practice for chronology of terrestrial rocks), but if Day et al. provide evidence why that is not applicable, then would a statement like this suffice at Line 337? “The nakhlites and chassignites have yielded ages ranging from ca. 1300 to 1400 Ma (Table S1). For the purposes of calculating the initial Sr and Nd isotopic compositions of these meteorites, we used an average of all of the ages for these meteorites, of 1340 ± 40 Ma.”

Elsewhere in the manuscript, the age of the nakhlites and chassignites should only appear with some indication of uncertainty (e.g., they should not be cited as an exact “1340 Ma”, which appears on lines 16, 61, 70, and 222). A possibility could be: “nakhlites and chassignites (ca. 1300-1400 Ma)”.

To avoid conflict regarding the ages of nakhlites, which is a contentious issue, we have removed figure S2 and simply cite the pertinent references for the ages. The mean of all ages, taken without prejudice, is 1340 +/- 40 Ma (Udry and Day, 2018) – we use this PUBLISHED value throughout the paper. We thank the reviewer for their view on these ages.

Various figures (especially 2-5):

Initial review: The chassignites and nakhlites are not identical, so it would be beneficial if these rock types had different symbols.

Rebuttal: We disagree – the whole point of this manuscript is to demonstrate that they are linked by source melting. We have kept the symbols as originally intended; this is also important for the sake of clarity.

Response to rebuttal: Day et al are indeed correct that the chassignites and nakhlites are very similar chemically, and I agree with their interpretation in this manuscript that they are linked by source melting. However, I will re-iterate my earlier point that these rocktypes are not identical - that’s why they have been classified into different groups. In particular, the nakhlites have abundant clinopyroxene, while the chassignites are rich in olivine. These mineralogical differences will affect the geochemistry (e.g., especially MgO on Figure 3). I think that the point that the nakhlites and chassignites are linked by source melting would actually be enhanced by having (subtly) different symbols for the two groups. A very easy option would be to keep using stars with a black outlines as the symbol for both the chassignites and the nakhlites. To distinguish the chassignites from the nakhlites, one of these groups (possibly the chassignites, as there are fewer of them) could

have the inside of the star shaded a different colour. This option would not compromise the clarity of the figures, but would enhance the science.

The chassignites are clearly demarked in the figure caption of the revised manuscript. No change – this is author preference and supported by our recent work (Udry & Day, 2018)

Figure 2: the blue dots should also be explained, either in the legend or the caption.

They were, and are now even more clearly, in the revised version.

Figure 4: the caption needs the statement “Analytical uncertainties are smaller than symbols”.

It’s a logarithmic scale; added.

Figure 5: the grey dashed lines still need to be explained.

Added.

Table S1: ages are listed for the shergottites, but no sources for these ages are reported.

The source of the ages was clearly reported in the reviewed manuscript as Ref. 19. No change.

Table S1: it isn't immediately apparent why are there three columns for the Total Analytical Blanks. There needs to be some more explanation of this aspect of the data.
Individual blanks are now reported.

Table S1 and S2: the supplementary files provided for this review were in the format of pdf portrait pages. This resulted in two major issues: 1) all pages except the first page lacked the column on the left that describes what was on each row, and 2) any text on the header and footer lines were split over multiple pages. This may be a formatting issue that has occurred during the journal electronic submission process, but these issues are obviously not ideal for readability, and need to be resolved before publishing.

Thank you. Providing excel spreadsheets would be the right thing to do during review, but this is a publisher issue. We have checked the tables very carefully.

Comments of Reviewer 2

This is the second round of reviews, but for Nat Comm, which allows for longer papers. I think the authors have answered and, in some cases, rebutted sensibly the reviewer comments on the first version. The paper is interesting and presents an important unifying concept, put into a global perspective. It puts the compositions of Martian meteorites into the framework of a plume-driven planet, which is consistent with geological observations and geophysical modeling. I predict that it will be widely cited and will be the springboard for numerous studies. Every new Martian igneous rock we analyze will be put into the same framework to see if it fits.

Thank you to reviewer 2 for these comments.

Comments of Reviewer 3, Anonymous

This manuscript, which provides new chemical data and a novel petrogenetic view of the Martian meteorite (SNC) suite is much improved over the first round. Below I list my main comments from last time and how the authors have responded.

Thank you.

1. TERRESTRIAL SURFACE PROCESSES AFFECTING METEORITES (sufficiently addressed):

Spikes in fluid-mobile elements (Ba, U, Sr) in a few of depleted shergottite patterns are suggestive of terrestrial weathering or alteration. However, the authors now better explain why the HFSE should be relatively invulnerable to fluid-mediated alteration and weathering.

Addition of solid material was not considered: Contamination by "fluid-immobile" elements is still possible however if water infiltrating among cracks brings terrestrial dust with it, i.e., not dissolved, but in fine suspension. It would be reassuring to see that possibility excluded by using the trace element data set, possibly combined with isotopic data. One could, e.g., take the most depleted sample in each group and model the addition of a loess-like component to such that the maximum HFSE concentrations are reached. If either the resulting trace element pattern or isotope composition does not fit those of the actual high-HFSE samples, then possible contamination by terrestrial dust can be

eliminated. In short, it should be easy to test—with existing data—for the presence of a terrestrial dust contaminant. I think the authors can decide if they want to do this or not.

We saw no evidence for any solid material addition, and no reason to include this modelling. We do add a statement about the possibility of solid (we presume wind-blown dust) to samples – further, the evidence is simply not in the dataset. A key question is how one set of samples (shergottites) would end up with a completely different HFSE signature to another (nakhlites/chassignites) if some of these stones have been sitting in the same hot desert environment.

2. ANALYTICAL METHODS: SAMPLE DIGESTION (not sufficiently addressed)

In my previous review I had mentioned that hot-plate digestions often do not digest refractory HFSE-bearing minerals. If there is a systematic difference in the types or amounts of refractory minerals present in nakhlites and chassignites vs. shergottites, then hot-plate digestions might cause a bias.

I suspect that this is not the case with the current dataset, but it would be easy enough to test by doing high-pressure digestions (autoclave bombs) of just a few samples and comparing their positions to hot-plate digestions in Fig. 5.

This is not so easy, given the limited mass of samples. Nonetheless, please see the new section where we have addressed this issue.

In the rebuttal, the authors state they do digestions “with confidence” adding in line 250 of the revised manuscript: “Acid attack led to complete dissolution of rock samples to generate clear solutions, with no remaining solid material.” - I am sceptical that Ca+Mg-bearing samples went into complete solution in a 4:1 HF-HNO₃ mixture at the given sample-to-acid ratios. Also, simply taking up the samples in concentrated HNO₃ will generally not destroy fluorides, but several dry-downs with HNO₃ might work. Is that what was done? In summary think the authors should provide solid evidence that the hotplate digestions were effective, especially for refractory Zr-rich minerals, which would affect their Nb/Y vs. Zr/Y diagram.

The quality of our digestions and analysis are first rate and we address this point in the manuscript. We are absolutely certain that 100% digestion took place and we state as much clearly in the manuscript, citing strong evidence in support.

3. PARTIAL METLING MODEL (not sufficiently addressed):

The authors have greatly improved their explanation of the modelling in the methods. However, Figure 5a is still a bit cluttered (black model lines obscured by data), and the authors still do not list specific partition coefficients used in their model. Line 287: Citing a partition coefficient database is not helpful unless specific data are cited. More useful to those trying to reproduce the models here would be a simple table of the actual partition coefficients (or ranges thereof) that were actually used and their specific publication reference. The reason is that among different studies, absolute partition coefficients tend to vary much more than RATIOS of partition coefficients; it would be good to be able to assess (through the table) whether the authors have used absolute partition coefficients that are internally consistent with ratios thereof.

We have worked on the model to ensure the partition coefficients are cited adequately.

4. REJUVENATED VOLCANISM ON MARS (sufficiently addressed):

Last time I had wondered whether this idea, supported by Nb/Y vs. Zr/Y and qualitative isotopic arguments, could benefit from quantitative modelling using existing isotope data. I had meant whether the implied sources of shergottites (S) and nakhlites (N) plus chassignites (C) could be linked by not only trace element patterns, but also by the isotopic evolution of their sources. OR perhaps one could predict what the isotope compositions of shergottites extracted from the N+C source would be. The former is probably difficult however given the age gap between S and N+C, and both are probably hampered by unknowns regarding the metasomatic agent. For this reason, I think it is fine for the authors keep isotopic classifications qualitative (e.g., long-term depleted vs. enriched sources) as they currently stand.

Thanks.

Line-by-line specific comments; (*) indicates comments repeated from first review.

18: “lithology” means “the study of rocks,” and thus should not be used to indicate e.g., “rock type” or “lithologic unit,” even though such usage is rampant in the literature. I recommend changing “distinct lithologies” to “distinct rock types” or “distinct rock lineages”

Modified.

21(*): “lower coupled Nb/Y-Zr/Y” - as I alluded to in my previous review, this wording may be compact, but is vague. What is meant here is “lower Nb/Y at a given Zr/Y.” As written, a reader might think that shergottites are simply further down toward the Y-intercept on the same trend and N+C.

Changed – we appreciate clarity from the reviewer on this occasion.

21: “Shergottites have lower coupled Nb/Y-Zr/Y than nakhlites or chassignites, a trend that is nearly identical to Hawaiian main shield and rejuvenated volcanism on Earth.” I recommend changing “trend” to “pattern” (because there are two trends, not one) AND adding “, respectively,” after volcanism.

Changed

35(*) “Mars..experienced” - avoid anthropomorphisms. “undergone” or “had”

Changed

41: “due to” (adjectival) -> “owing to” (adverbial; modifying “have not been recognised”)

Okay

59: Figure 1: Just an observation: These samples are wound back to different times in the past, reflecting instantaneous isotopic snapshots of sources that evolved through time. As such, the figure blurs the time dimension so relations of among source reservoirs at any given point in time are difficult to make from this diagram. It does serve its purpose however in sorting sources into broadly depleted or enriched sectors.

This is correct. Portraying the isotopology of these various samples isn't easy. Modern sources is one option and it doesn't skew the relationship seen that much. It serves its purpose; the variation on Earth is much much smaller than in Mars.

64: "nakhlite and chassignites" (add an "s" to the former)

Added

68 Delete hyphen between late and incompatible.

Done.

82: "Mars experiences" - anthropomorphism. Reword.

Done.

86: "compare these data with THOSE obtained for nakhlites and chassignites.."

Fixed

114: "shergottites and nakhlites and chassignites" - I think in response to my earlier comment, a slash was replaced by an "and" but the grouping is not clear. Perhaps replace the first "and" with "vs."

Versus added. We appreciate this clarification.

117: add "s" to "nakhlite"

Done

119: again eliminating a slash made the grouping unclear. Suggest changing the second "and" to "plus" or similar.

Versus used again.

139-150: Inter-element ratios are "unaffected" by the mode effect. Define "unaffected" — no difference outside of analytical uncertainty? Within X%? Also: Would it make sense to use the mean values of replicate analyses to gain a more representative value?

Quantification has been added.

140: "meteoritics" - Do not make nouns out of adjectives by adding "s" (meteorites)

Typo – changed.

141: "due to" -> "owing to" (adjectival vs adverbial modifier; the verb here being "is")

Modified.

181(*) "pre-/post-shield stage" - avoid such vague slash constructions.

Modified.

187: "based on" (adjectival modifier) —> "on the basis of" (adverbial modifier), the modified verb being "are observed"

Modified.

188: “similar TO — although significantly larger than— flexural bulges” (add “to” and offsetting m-dashes or commas)

Modified.

201: “Models of 5-10% partial melting of primitive mantle reproduce Hawaiian shield compositions, and of depleted mantle reproduce mid-ocean ridge basalt compositions” - awkward sentence structure.. perhaps replace “and of depleted” with “whereas those of depleted”

Done.

214: is “tectonic” the right word here? This is more of a deeper geodynamic difference, right? Stirring and tectonic features are the results of convection. Convection and stirring occur in both plate tectonic and stagnant lid scenarios. Some studies, however, e.g., Debaille et al. (2013, EPSL 273), show that stagnant lid scenarios may better preserve isotopic variability than plate tectonic ones.

We feel it is the correct word.

222 “shergottite crystallization ages (165-2400 Ma) generally preclude a direct link between nakhlites and chassignites.” In the previous version of the manuscript, a direct link between younger shergottites and nakhlites was precluded by their age differences, which made sense to me. Now the precluded link is between nakhlites and chassignites — on the basis of shergottite ages, which does not make sense to me. Perhaps the authors meant: “..generally preclude a direct link to nakhlites and chassignites.”

We felt this was corrected adequately in the first revision – we have modified.

224(*): “from a source that has seen prior melt depletion” - sources do not “see” anything. Avoid anthropomorphism.

Done.

226: Observations of 2370 to 2400 Ma shergottite magmatism on Mars supports continued and persistent plume magmatism for at least two billion years” — Well the ages quoted here just mark the older end of the magmatism interval. Perhaps quote and cite the full shergottite age range here.

Done.

231: “The distinctive nature of martian meteorites [as compared to] remotely sensed martian surface samples” OR “The distinctive natures of martian meteorites [and] remotely sensed martian surface samples”

We prefer our wording.

235: “offering a mechanism for explaining” = “potentially explaining”

We prefer our wording.

279: Consider rewording: “correction for non-systematic laboratory bias” - A bias is a systematic error. Biases between labs can be corrected using data from international standards published alongside the unknowns. What cannot be corrected are “non-

systematic” (random) errors within a lab’s data. Replicate data of standards from a single laboratory give the external reproducibility and thus is a measure of the magnitude of the random error in a lab’s data.

Done.

284: “the source[s] of rejuvenated lavas and nakhlite[s] and chassignites versus [the sources] of Hawaiian main shield stage lavas and shergottites.

Done

337: “based on” -> “on the basis of”

Done

Table S1:

There are still concentrations and blanks listed in this table that have 4 and 5 significant digits, which is unrealistically precise given the external reproducibility.

We report the significant figures according to the total analytical blank and to the % external uncertainty. No value in this table is ‘over reported’.

Table S2:

Move the method column next to the crystallisation ages. The given methods are not for exposure ages.

Changed

“standard deviation” is usually abbreviated as S.D., SD, or s.d.

St. Dev.

Reviewers' Comments:

Reviewer #1:

Remarks to the Author:

Please see attached pdf.

Reviewer #3:

Remarks to the Author:

(Reviewer 3; see my previous reviews)

In their latest revision of their article, the authors have fully addressed my remaining concerns from the second draft. They now demonstrate that their digestion technique is sufficient to reproduce critical element ratios to within 2% and they have listed the exact partition coefficients used in their modelling. This is a very interesting paper that presents a valuable dataset and a novel and thought-provoking model linking the petrogenesis of shergottite, nakhlite and chassignite magmas.

My remaining comments deal with mostly improving the wording of a few passages and correcting a typo:

138-141: "Terrestrial alteration has had no effect on the majority of the rare earth elements (REE) or high field strength elements (HFSE) such as Nb, Ta, Hf, Zr or Y, however, with well-defined groupings for nakhlites and chassignites, and for incompatible-element enriched, intermediate and depleted shergottites for Nb/Y, La/Yb, and Zr/Nb (Figure 3)." — This should be split into two sentences.

186: Perhaps explain "subsurface loads" for a wider audience

212-214: "Low degree partial melts of a primitive mantle source could, in theory, generate Hawaiian rejuvenated lavas, but is inconsistent with their depleted source character from Sr-Nd isotope systematics." — "melts" is plural, so maybe change "is" to "are" or to "this is" (referring to the idea rather than melts).

220: "convective mixing to homogenize mantle isotopic compositions " — the "to" implies purpose. There is no purpose here, just cause and effect. So: I suggest rewording, for example: "...would suggest that the homogenisation of mantle isotopic compositions by convective mixing has not been as effective..."

236-238: "Observations of 2370 to 2400 Ma shergottite magmatism on Mars supports continued and persistent plume magmatism for at least two billion years." — this statement is only true when one ALSO considers observations of the "young" shergottites as well.

262-263: "Acid attack led to complete dissolution of rock samples to generate clear solutions, with no remaining solid material." - Again the "to" (before generate) implies a purpose, rather than describing cause and effect: I suggest "Acid attack led to complete dissolution of rock samples, generating clear solutions, with no remaining solid material."

298: "We no observable differences," - Correct this typo.

Fig 5. I think that the figure could be made clearer (for a wider audience) by explaining the dashed lines between the garnet and spinel lines.

I hope the authors find my comments helpful.

The manuscript has been improved, however there are still some outstanding issues, outlined below.

Lines 106-108: for this statement about NWA 6963 having a ‘major-element composition of a shergottite’, the authors need to support this statement with appropriate evidence. Figure 2 is not suitable because the fields for shergottites (red outline) and nakhlites (black outline) largely overlap on Figure 2 (below). In fact, on Figure 2, the chemistry of NWA 6963 (red arrow) is actually most similar to a *nakhlite* (black arrow), not another shergottite.

Line 277: the text ‘this method’ is somewhat ambiguous. Could the authors clarify by writing the specific method, e.g., ‘ICP-MS methodology for determination of major- and trace-....’

Line 294: what is meant by ‘(e.g., Nakhla)’?

Line 247-248 and Table 1: the authors also need to state that the geochemical data for the nakhlite and chassignite meteorites are from Udry & Day (2018).

Lines 258-259: This text is an improvement, but the exact mass crushed for each sample would be a valuable addition to Table 1 (my apologies, the previous review should ideally have also specified ‘Table 1’ rather than just ‘Methods’). This is because the exact mass crushed for each sample is important for quality assessment and data traceability. For example, a sample with only 0.5 grams crushed will be more susceptible to the mode effect compared to a sample with 2 grams crushed.

Figures 2 and 3, the captions have been modified to state that:

Figure 2: ‘Nakhlite data includes chassignite samples that fall below 41 wt.% SiO₂.’

Figure 3: ‘MgO versus Ce/Pb, Nb/Y, La/Yb, Ba/Nb, Zr/Ti and Zr/Nb for shergottites, nakhlites and chassignites (>30 wt.% MgO).’

I think that the authors are trying to say that the chassignite samples are the star symbols with < 41 wt.% SiO_2 and $>30\%$ MgO. The current sentence structures are, however, ambiguous, e.g., for the Figure 3 caption does the text ' >30 wt % MgO' refer to just the chassignites, the nakhlites and chassignites, or all of the rocktypes listed?

These captions could be improved by writing:

Figure 2: ' (Note the chassignite meteorites are the star symbols with < 41 wt.% SiO_2).'

Figure 3: 'shergottites, nakhlites, and chassignites. (Note the chassignite meteorites are the star symbols with $>30\%$ MgO).'

Regarding the ages for the nakhlite and chassignite meteorites, I agree that removing Figure S2 is an acceptable way forward for this manuscript. In their rebuttal, the authors state that the number of 1340 ± 40 Ma (used on lines 62-63) is 'published', and that it is the 'mean of all ages, taken without prejudice'. However, use of a single average age \pm standard deviation, even if published, is unfortunately not a statistically valid or geologically useful treatment of analytically distinct chronologic ages. Statistical measures for the nakhlite ages (using the numbers reported in Table S2 in the second submitted version) yields a MSWD of 5.8 and p of 0.00. For statistically valid average ages, values for MSWD and p should be <2 and >0.05 , respectively; values outside these ranges indicate that scatter exceeds the analytical uncertainties (Renne et al. 2009. *Quat. Geochron.* 4, 346-352.).

As an analogy, consider a hypothetical volcano with ages of 2 ± 0.1 Ma, 1.5 ± 0.1 Ma, 1.2 ± 0.05 Ma, 0.8 ± 0.05 Ma, 0.5 ± 0.05 Ma, and 0 ± 0 Ma (i.e., an eruption currently underway). The average is 1 ± 0.7 Ma (1 sigma, MSWD = 300, p = 0.00). Geologically, the average of 1 ± 0.7 Ma does not reflect the fact that there were 6 eruptions, that eruptions spanned over 2 Ma, or that the individual eruptions had much more precise ages. Furthermore, text like on line 71, 'crystallization of the nakhlites at ~ 1340 Ma' is analogous to saying 'the volcano is ~ 1 Ma old' – which is an incorrect statement for a volcano with an active eruption.

In this paper, at lines 62-63, it would be reasonable for Day et al. to say something like 'chronologic ages for the nakhlites and chassignites cluster in the 1300-1400 Ma range (References)'. That is a statement of facts, and has less prejudice or interpretation than calculating an average age. Similarly, at line 71, they could use text like 'and the ~ 1300 -1400 Ma crystallization of the nakhlites and chassignites'.

Referee 1

The manuscript has been improved, however there are still some outstanding issues, outlined below.

Lines 106-108: for this statement about NWA 6963 having a ‘major-element composition of a shergottite’, the authors need to support this statement with appropriate evidence. Figure 2 is not suitable because the fields for shergottites (red outline) and nakhlites (black outline) largely overlap on Figure 2 (below). In fact, on Figure 2, the chemistry of NWA 6963 (red arrow) is actually most similar to a *nakhlite* (black arrow), not another shergottite.

Modified. We do not, however, think that due to differentiation, the proximity of a sample on a TAS plot does not infer similarity. Examples would be differentiation trends in terrestrial volcanic systems that superimpose but do not suggest sample similarity. Figure 2 is entirely appropriate for showing the data in the format described.

Line 277: the text ‘this method’ is somewhat ambiguous. Could the authors clarify by writing the specific method, e.g., ‘ICP-MS methodology for determination of major and trace-....

Modified.

Line 294: what is meant by ‘(e.g., Nakhla)’?

Modified.

Line 247-248 and Table 1: the authors also need to state that the geochemical data for the nakhlite and chassignite meteorites are from Udry & Day (2018).

Modified.

Lines 258-259: This text is an improvement, but the exact mass crushed for each sample would be a valuable addition to Table 1 (my apologies, the previous review should ideally have also specified 'Table 1' rather than just 'Methods'). This is because the exact mass crushed for each sample is important for quality assessment and data traceability. For example, a sample with only 0.5 grams crushed will be more susceptible to the mode effect compared to a sample with 2 grams crushed.

Added to the table.

Figures 2 and 3, the captions have been modified to state that:

Figure 2: 'Nakhlite data includes chassignite samples that fall below 41 wt.% SiO₂.'

Figure 3: 'MgO versus Ce/Pb, Nb/Y, La/Yb, Ba/Nb, Zr/Ti and Zr/Nb for shergottites, nakhlites and chassignites (>30 wt.% MgO).'

I think that the authors are trying to say that the chassignite samples are the star symbols with < 41 wt.% SiO₂ and >30% MgO. The current sentence structures are, however, ambiguous, e.g., for the Figure 3 caption does the text '>30 wt % MgO' refer to just the chassignites, the nakhlites and chassignites, or all of the rocktypes listed?

These captions could be improved by writing:

Figure 2: '(Note the chassignite meteorites are the star symbols with < 41 wt.% SiO₂).'

Figure 3: 'shergottites, nakhlites, and chassignites. (Note the chassignite meteorites are the star symbols with >30% MgO).'

Modified, with thanks.

Regarding the ages for the nakhlite and chassignite meteorites, I agree that removing Figure S2 is an acceptable way forward for this manuscript. In their rebuttal, the authors state that the number of 1340 ± 40 Ma (used on lines 62-63) is 'published', and that it is the 'mean of all ages, taken without prejudice'. However, use of a single average age \pm standard deviation, even if published, is unfortunately not a statistically valid or geologically useful treatment of analytically distinct chronologic ages. Statistical measures for the nakhlite ages (using the numbers reported in Table S2 in the second submitted version) yields a MSWD of 5.8 and p of 0.00. For statistically valid average ages, values for MSWD and p should be <2 and >0.05, respectively; values outside these ranges indicate that scatter exceeds the analytical uncertainties (Renne et al. 2009. *Quat. Geochron.* 4, 346-352.). As an analogy, consider a hypothetical volcano with ages of 2 ± 0.1 Ma, 1.5 ± 0.1 Ma, 1.2 ± 0.05 Ma, 0.8 ± 0.05 Ma, 0.5 ± 0.05 Ma, and 0 ± 0 Ma (i.e., an eruption currently underway). The average is 1 ± 0.7 Ma (1 sigma, MSWD = 300, p = 0.00). Geologically, the average of 1 ± 0.7 Ma does not reflect the fact that there were 6 eruptions, that eruptions spanned over 2 Ma, or that the individual eruptions had much more precise ages. Furthermore, text like on line 71, 'crystallization of the nakhlites at ~1340 Ma' is analogous to saying 'the volcano is ~ 1 Ma old' – which is an incorrect statement for a volcano with an active eruption. In this paper, at lines 62-63, it would be reasonable for Day et al. to say something like 'chronologic ages for the nakhlites and chassignites cluster in the 1300-1400 Ma range (References)'. That is a statement of facts, and has less prejudice or interpretation than calculating an average age. Similarly, at line 71, they could use text like 'and the ~1300-1400 Ma crystallization of the nakhlites and chassignites'.

We thank the reviewer for being explicit in their point here and we agree and modify accordingly.

Referee 3

In their latest revision of their article, the authors have fully addressed my remaining concerns from the second draft. They now demonstrate that their digestion technique is sufficient to reproduce critical element ratios to within 2% and they have listed the exact partition coefficients used in their modelling. This is a very interesting paper that presents a valuable dataset and a novel and thought-provoking model linking the petrogenesis of shergottite, nakhlite and chassignite magmas.

We thank the reviewer for taking the time to read our paper again and to provide further useful comments.

My remaining comments deal with mostly improving the wording of a few passages and correcting a typo:

138-141: “Terrestrial alteration has had no effect on the majority of the rare earth elements (REE) or high field strength elements (HFSE) such as Nb, Ta, Hf, Zr or Y, however, with well-defined groupings for nakhlites and chassignites, and for incompatible-element enriched, intermediate and depleted shergottites for Nb/Y, La/Yb, and Zr/Nb (Figure 3).” — This should be split into two sentences.

Agreed – done.

186: Perhaps explain “subsurface loads” for a wider audience

Modified.

212-214: “Low degree partial melts of a primitive mantle source could, in theory, generate Hawaiian rejuvenated lavas, but is inconsistent with their depleted source character from Sr-Nd isotope systematics.” — “melts” is plural, so maybe change “is” to “are” or to “this is” (referring to the idea rather than melts).

Modified.

220: “convective mixing to homogenize mantle isotopic compositions “ — the “to” implies purpose. There is no purpose here, just cause and effect. So: I suggest rewording, for example: “...would suggest that the homogenisation of mantle isotopic compositions by convective mixing has not been as effective...”

Modified.

236-238: “Observations of 2370 to 2400 Ma shergottite magmatism on Mars supports continued and persistent plume magmatism for at least two billion years.” — this statement is only true when one ALSO considers observations of the "young" shergottites as well.

Modified.

262-263: “Acid attack led to complete dissolution of rock samples to generate clear solutions, with no remaining solid material.” - Again the “to” (before generate) implies a purpose, rather than describing cause and effect: I suggest “Acid attack led to complete dissolution of rock samples, generating clear solutions, with no remaining solid material.”

Modified.

298: “We no observable differences,” - Correct this typo.

Modified.

Fig 5. I think that the figure could be made clearer (for a wider audience) by explaining the dashed lines between the garnet and spinel lines.

Modified.